# Extracellular Proteolytic Activity and Amino Acid Production by Lactic Acid Bacteria Isolated from Malaysian Foods

**DOI:** 10.3390/ijms20071777

**Published:** 2019-04-10

**Authors:** Cui Jin Toe, Hooi Ling Foo, Teck Chwen Loh, Rosfarizan Mohamad, Raha Abdul Rahim, Zulkifli Idrus

**Affiliations:** 1Department of Bioprocess Technology, Faculty of Biotechnology and Biomolecular Sciences, Universiti Putra Malaysia, 43400 UPM Serdang, Selangor, Malaysia; tcjin0314@hotmail.com (C.J.T.); farizan@upm.edu.my (R.M.); 2Institute of Bioscience, Universiti Putra Malaysia, 43400 UPM Serdang, Selangor, Malaysia; raha@upm.edu.my; 3Department of Animal Science, Faculty of Agriculture, Universiti Putra Malaysia, 43400 UPM Serdang, Selangor, Malaysia; zulidrus@upm.edu.my; 4Institute of Tropical Agriculture and Food Security, Universiti Putra Malaysia, 43400 UPM Serdang, Selangor, Malaysia; 5Institute of Tropical Forestry and Forest Products, Universiti Putra Malaysia, 43400 UPM Serdang, Selangor, Malaysia; 6Department of Cell and Molecular Biology, Faculty of Biotechnology and Biomolecular Sciences, Universiti Putra Malaysia, 43400 UPM Serdang, Selangor, Malaysia; 7Halal Products Research Institute, Universiti Putra Malaysia, 43400 UPM Serdang, Selangor, Malaysia

**Keywords:** amino acid, lactic acid bacteria, *Pediococcus pentosaceus*, *Pediococcus acidilactici*, *Lactobacillus plantarum*, extracellular proteolytic activity

## Abstract

Amino acids (AAs) are vital elements for growth, reproduction, and maintenance of organisms. Current technology uses genetically engineered microorganisms for AAs production, which has urged the search for a safer food-grade AA producer strain. The extracellular proteolytic activities of lactic acid bacteria (LAB) can be a vital tool to hydrolyze extracellular protein molecules into free AAs, thereby exhibiting great potential for functional AA production. In this study, eight LAB isolated from Malaysian foods were determined for their extracellular proteolytic activities and their capability of producing AAs. All studied LAB exhibited versatile extracellular proteolytic activities from acidic to alkaline pH conditions. In comparison, *Pediococcus pentosaceus* UP-2 exhibited the highest ability to produce 15 AAs extracellularly, including aspartate, lysine, methionine, threonine, isoleucine, glutamate, proline, alanine, valine, leucine, tryptophan, tyrosine, serine, glycine, and cystine, followed by *Pediococcus pentosaceus* UL-2, *Pediococcus acidilactici* UB-6, and *Pediococcus acidilactici* UP-1 with 11 to 12 different AAs production detected extracellularly. *Pediococcus pentosaceus* UL-6 demonstrated the highest increment of proline production at 24 h of incubation. However, *Pediococcus*
*acidilactici* UL-3 and *Lactobacillus plantarum* I-UL4 exhibited the greatest requirement for AA. The results of this study showed that different LAB possess different extracellular proteolytic activities and potentials as extracellular AA producers.

## 1. Introduction

Amino acids (AAs) are amphoteric acids composed of amine and carboxylic acids as functional groups [1]. They are the building blocks of proteins, which are important in sustaining the growth of living organisms [2]. The present study focuses on a new concept of functional AAs, which is defined as AAs that regulate key metabolic pathways [3]. Deficiency in any of the functional AAs in animals could halt protein biosynthesis in the animal and cause severe growth impairments [4,5]. A study conducted by Novak et al. [6] concluded that the combination of dietary lysine and sulfur containing AAs (cysteine and methionine) could improve the egg quality and production of laying hens. Moreover, AAs in nutrition also affects the whole-body homeostasis and helps in multiple regulatory functions in cells [7]. Currently, the extensive use of AAs in feed industries has gained considerable interest. Supplementation of AAs complements the inadequate amount of limiting AAs in feedstuff, thus increasing the nutritional value of feed, which improves the performance and well-being of animals [8,9]. Moreover, the increase in the human population has led to tremendous escalation of meat demand, which has promoted the expansion of the livestock industry and elevated the demand of AAs as feed supplement. 

Given the high demand of AAs, production of AAs via sustainable alternative microbial approach have gained marked interest in the industry [10]. Production of AAs by microbial approach can be further divided into two different processes, namely fermentation and enzymatic methods [11]. Nowadays, fermentation is widely used in the industry for the production of most AAs, except achiral AA glycine and chiral AA methionine, which are preferably synthesized chemically. Fermentative production of AAs is favorable because it is one of the most economical ways to produce l-amino acids in bulk. Genetically engineered *Corynebacterium glutamicum* is the dominating microorganism for commercial AA production at present [12]. It was first isolated in Japan by Kinoshita et al. [13] with a remarkable ability to achieve up to 26 g/L of glutamate production in minimal medium under biotin-limited conditions after mutation under UV radiation. Subsequently, other AAs including lysine, histidine, isoleucine, tryptophan, phenylalanine, valine, alanine, serine, and arginine were also produced by the genetically modified bacteria [14]. The use of genetically engineered microorganisms that are not food grade has been a major concern in food and feed industries. This has led to the search of safer food-grade producer strains. Recent studies showed that wild type lactic acid bacteria (LAB) have the potential to be considered as one of the safer candidates for the production of AAs [15,16]. 

LAB are characterized as Gram positive, non-sporulating Firmicutes, and they produce lactic acid as a major end product through fermentative metabolism of carbohydrates [17]. They have been extensively studied and employed in various industries because of their economic advantages and generally recognized as safe (GRAS) reputation. For instance, LAB are used extensively as starter cultures in the production of dairy fermented foods and beverages [18,19]. One of the distinguished characteristics of LAB is their fastidious nutrient requirements, especially in nitrogen sources [20]. LAB propagate poorly in an environment that contains inorganic nitrogen as the sole nitrogen source. They often require an exogenous supply of peptides and AAs for survival and growth [21]. Hence, a majority of LAB possess a well-established proteolytic system with complex combinations of proteinases and peptidases to obtain AAs from complex peptides [22]. Many studies have concluded that the proteolytic system of LAB is important in both utilization of protein and peptides for growth and in maturation process of milk in dairy products [23,24,25,26]. Gobbetti et al. [27] showed that proteinase activity was affected by pH in addition to the effects of substrate and bacterial strains. The elaborate proteolytic system of LAB is a prerequisite to breakdown complex polypeptides and supplies the bacteria with AAs to ensure their survival [28]. By exploiting the proteolytic system of LAB, protein molecules could be hydrolyzed into peptides and free AAs in vitro, which can be used as health supplements and feed supplements in animal feeding [29]. 

Although extensive reports on proteolytic activity of LAB are available, the documentation on the application of LAB to produce extracellular AAs is very limited. Hence, the objectives of this study were to explore the extracellular proteolytic activity of eight LAB isolated from various Malaysian foods and to evaluate their ability to produce extracellular AAs. 

## 2. Results and Discussion

### 2.1. Proteolytic Activity of Lactic Acid Bacteria (LAB)

In this study the effects of pH on extracellular proteolytic activities of eight LAB isolated from Malaysian food were investigated. The extracellular proteolytic activity was quantified spectrophotometrically using azocasein as a substrate. Proteolytic enzymes cleaved azocasein and released short peptides containing an azo compound, which resulted in the development of a yellow color. The specific proteolytic activities of LAB isolates in three different pH conditions are shown in Figure 1. All the LAB isolates showed proteolytic activity towards azocasein, indicating that the LAB isolates were able to release short peptides from azocasein. Among the eight studied LAB isolates, five LAB isolates demonstrated the highest specific extracellular proteolytic activity in alkaline conditions (pH 8), while the remaining three LAB isolates produced the highest specific extracellular proteolytic activity in acidic conditions (pH 5), hence, suggesting that the majority of the tested LAB produced proteolytic enzymes that were most active in alkaline conditions, although they grew well in acidic conditions. The results of the proteolytic activities were in agreement with two studies conducted by Dalmis and Soyer [30] and Essid et al. [31], who demonstrated that alkaline proteases were also produced by their studied LAB. 

On the other hand, *Pediococcus pentosaceus* UB-8 achieved the highest specific proteolytic activity in acidic conditions, but it was not significantly different (*p* > 0.05) as compared to *Pediococcus pentosaceus* UP-2. Meanwhile, *Pediococcus acidilactici* UL-3 achieved the highest specific proteolytic activity in near neutral and alkaline conditions. Gobbetti et al. [27] showed that proteinase activity was affected by pH and was often substrate- and strain-dependent. Detection of proteolytic activity in three different pH conditions suggested that the studied LAB produced more than one type of extracellular proteolytic enzyme, which were versatile and active over broad pH conditions ranging from acidic to alkaline. This is in line with the findings reported by Simitsopoulou et al. [32], whereby *Pediococcus* sp. isolated from cheddar and feta cheeses possessed aminopeptidase, dipeptidase, dipeptidyl aminopeptidase, and protease activities. Fadda et al. also showed that *Lactobacillus sakei*, *Lactobacillus curvatus*, and *Lactobacillus plantarum* isolated from sausages possessed several peptidase activities [33,34]. Presence of proteolytic activity in all the studied LAB may contribute to AA production. Therefore, their ability to produce AAs was investigated in the subsequent experiment.

### 2.2. Production of Amino Acids

#### 2.2.1. Cell Population and Reducing Sugar Utilization

Each LAB isolate manifested exponential growth in the first 8 h of incubation (Figure 2A–H) except for *Pediococcus acidilactici* UP-1, *P. pentosaceus* UP-2, and *P. acidilactici* UL-3, whereby their growth was slower compared to the other LAB isolates and reached a stationary phase in the later part of the incubation. The stationary phase of UP-1 started at 12 h of incubation, UP-2 at 16 h of incubation, and UL-3 at 20 h of incubation. Only 10 g/L of the reducing sugar was consumed by *P. acidilactici* UP-1, *P. pentosaceus* UP-2, and *P. acidilactici* UL-3 after 24 h of incubation. The remaining reducing sugar in the medium was relatively high in comparison to the other tested LAB, indicating that *P. acidilactici* UP-1, *P. pentosaceus* UP-2, and *P. acidilactici* UL-3 utilized reducing sugar at a slower rate and reached the stationary phase slightly later compared to other LAB isolates. *Pediococcus acidilactici* UB-6 utilized slightly more reducing sugar compared to *P. acidilactici* UP-1, *P. pentosaceus* UP-2, and *P. acidilactici* UL-3. Generally, the majority of the *P. pentosaceus* isolates consumed more reducing sugar compared to *P. acidilactici* isolates, except *P. pentosaceus* UP-2. All tested LAB showed a relatively high cell population at the stationary phase with an optical density at 600 nm (OD_600nm_) around 2. The highest cell growth was recorded in *L. plantarum* I-UL4 with an OD_600nm_ of 2.13 at 22 h. A similar trend was observed in the reducing sugar consumption profile, whereby *L. plantarum* I-UL4 consumed the highest amount of reducing sugar (23.91 g/L) in comparison to the other LAB isolates.

#### 2.2.2. Amino Acid Production Profile

Increments of isoleucine, proline, glutamate, and glycine content were detected in cell-free supernatant (CFS) of all the studied LAB isolates, suggesting that they had the ability to produce those AAs extracellularly (Table 1, Table 2, Table 3, Table 4, Table 5, Table 6, Table 7 and Table 8). A study conducted by Liu et al. [23] showed that all LAB genomes were encoded with many types of peptidases including the proline-specific peptidases PepX and PepQ. The proline-specific peptidases were specific towards proline dipeptides and they cleaved the peptide bonds of the dipeptides to release proline and a free AA. Therefore, high levels of proline in all the studied LAB isolates may be due to the presence of proline peptidase activity. Among all the studied LAB isolates, *P. pentosaceus* isolates exhibited a relatively higher percentage of proline production. *Pediococcus pentosaceus* UL-6 showed the highest percentage of proline increment, 121.88% at 24 h of incubation, followed by *P. pentosaceus* UB-8 (with approximately two folds lower proline production). A similar result was reported by Vafopoulou-Mastrojiannaki et al. [35], whereby *P. pentosaceus* strains exhibited high dipeptidyl aminopeptidase activity towards L-glycyl-proline-pNA, which cleave dipeptides with proline residue. *P. pentosaceus* UL-6 demonstrated the capability of producing isoleucine, glutamate, proline, valine, phenylalanine, glycine, and cysteine (Table 8). Meanwhile, *P. pentosaceus* UB-8 demonstrated the capability to produce aspartate, asparagine, arginine, alanine, and tryptophan (Table 4). Most of the AAs produced by *P. pentosaceus* UB-8 reached a maximum after 14 h of incubation. Although *P. pentosaceus* UL-6 and UB-8 were able to produce high amounts of proline, the production of other AAs was relatively low.

Out of the eight studied LAB isolates, *P. pentosaceus* UP-2 was the best AA producer because it was able to produce a total of 15 different AAs, including aspartate, lysine, methionine, threonine, isoleucine, glutamate, proline, alanine, valine, leucine, tryptophan, tyrosine, serine, glycine, and cystine (Table 2). Moreover, the amount of most of the AAs produced by *P. pentosaceus* UP-2, including glutamate, leucine, cysteine, alanine, isoleucine, threonine, serine, methionine, lysine, aspartate, tyrosine, and tryptophan, was the highest among all the tested LAB. Among the AAs produced, glutamate had the highest increment, which was 98.0 mg/L at 18 h of incubation, followed by leucine with 95.4 mg/L at 18 h of incubation. *P. pentosaceus* UP-2 exhibited a distinguished ability to produce lysine, with an amount equal to 50.8 mg/L. The amount was relatively low compared to industrial *Corynebacterium glutamicum* strains that have been genetically modified on defined genome-based, whereby the strains were able to produce up to 120 g/L of lysine [36]. A study carried out by Moosavi-Nasab et al. [37] showed that *C. glutamicum* (PTCC 1532) was able to produce up to 48 g/L of lysine at 96 h of incubation using molasses as a substrate. Although the production of lysine by *P. pentosaceus* UP-2 was lower than *C. glutamicum*, the ability of *P. pentosaceus* UP-2 to produce a variety of AAs has opened up a new application, particularly in the feed industry. Wendisch and Bott [38] suggested that lysine is one of the most important essential AAs used as supplement in animal feed, whereby supplementation of lysine has boosted the growth of chicken and pigs. Other than *P. pentosaceus* UP-2, *P. acidilactici* UB-6 also demonstrated the ability to produce lysine.

*P. acidilactici* UB-6, *P. acidilactici* UP-1, and *P. pentosaceus* UL-2 demonstrated a relatively high AA production capacity compared to other studied LAB isolates. For instance, *P. acidilactici* UB-6 had the ability to produce up to a total of 13 different AA. Chopin [39] suggested that LAB isolated from plants were prototrophic to most AAs. The AA production profile (Table 3) suggested that *P. acidilactici* UB-6 consumed the AAs that were present in de Man, Rogosa, and Sharpe (MRS) medium for its growth in the initial stage of the fermentation, whereby most of the AA concentration was reduced during the first 2 h of incubation. Production for most of the AAs by *P. acidilactici* UB-6 achieved maximum levels between 14 h to 20 h of incubation, indicating the production of AA by *P. acidilactici* UB-6 occurred in the later stage of fermentation. The AAs that produced were included lysine, methionine, threonine, isoleucine, glutamate, proline, alanine, valine, tryptophan, serine, and glycine. The most AA produced by *P. acidilactici* UB-6 was lysine (35.8 mg/L) at 18 h of incubation. On the other hand, arginine, phenylalanine, and tyrosine were completely utilized by *P. acidilactici* UB-6, implying that these AAs were crucial for the survival of the isolate. This is in agreement with the findings reported by Sriphochanart et al. [40], whereby phenylalanine and tyrosine were critical for the growth and lactic acid production by LAB.

*P. acidilactici* UP-1 (Table 1) and *P. pentosaceus* UL-2 (Table 5) were able to produce 11 different types of AA, respectively. Most of the AAs produced by *P. acidilactici* UP-1 achieved the highest level between 10 to 18 h of incubation. The maximum glutamate amount was detected at 10 h of incubation, which was 68.6 mg/L, followed by the production of isoleucine, which was 51.8 mg/mL. *P. acidilactici* UP-1 was also able to produce lysine, with an amount of 15.7 mg/L at 4 h of incubation. *P. acidilactici* UP-1 had the ability to produce the highest concentration of asparagine compared to other LAB isolates, with an amount of 30.3 mg/L. On the other hand, *P. pentosaceus* UL-2 produced the highest amount of leucine in comparison to the other 11 AAs. Production of leucine by *P. pentosaceus* UL-2 increased significantly from 0 h to 12 h with the highest concentration of 302.9 mg/L. Supplementation of leucine into a low-protein diet in weaning pigs increased tissue production synthesis [41]. In addition, leucine also affects cell signaling, proliferation, and migration [42]. A similar trend was observed in glutamate production by *P. pentosaceus* UL-2, whereby it reached the highest concentration at 10 h of incubation. Other AAs, such as threonine, glycine, and isoleucine, achieved the highest concentration at 16 h of incubation. From the AA production profile of *P. pentosaceus* UL-2, the production rate of most AAs was higher than the rate of utilization during the initial phase of fermentation (0–16 h), resulting in the increasing amount of AAs during this phase. In contrast, asparagine, tryptophan, aspartate, and arginine were utilized completely by *P. pentosaceus* UL-2 throughout the incubation period, inferring that these AAs were extremely crucial for the growth of *P. pentosaceus* UL-2. A study reported by Cunin et al. [43] demonstrated that some LAB were able to metabolize arginine as an energy source via arginine decarboxylase.

*P. acidilactici* UL-3 (Table 6) and *L. plantarum* I-UL4 (Table 7) demonstrated the highest AA requirement, whereby they possessed a limited ability to produce most of the AAs. For instance, *P. acidilactici* UL-3 was unable to produce threonine, arginine, tyrosine, tryptophan, phenylalanine, and lysine, whereby these AAs decreased with the incubation time. In addition, insignificant (*p* > 0.05) amounts of aspartate, asparagine, alanine, and cysteine were produced by *P. acidilactici* UL-3. In total, *P. acidilactici* UL-3 was only able to produce eight AAs, with the highest increment of proline at 87.8 mg/L. Meanwhile, *L. plantarum* I-UL4 exhibited the greatest requirement for free AA compared to the other tested LAB isolates. The result was similar to the findings of Morishita et al. [44], who demonstrated that *Lactobacilli* had the most extensive requirements for essential AAs. Although increments of some AAs could be observed at 6 h of incubation, the AA concentrations were decreased after 6 h of incubation, implying that these AAs were essential for the growth and survival of *L. plantarum* I-UL4, particularly during the exponential growth phase. Yvon et al. [45] reported that *L. plantarum* transaminated leucine, phenylalanine, and other AAs to other compounds. Decreasing amounts of leucine, phenylalanine, and other AAs in the growth medium of *L. plantarum* I-UL4 suggested that transamination may occur during the fermentation phase. Although a majority of the AAs exhibited a decreasing profile, an increasing glutamate concentration was detected in *L. plantarum* I-UL4. This might be due to the absence of glutamate dehydrogenase activity in *L. plantarum*. As a consequence, the *L. plantarum* was unable to metabolize glutamate [46].

#### 2.2.3. Extracellular Proteolytic Activity Profile of LAB

To the best of our understanding, only the proteolytic activity of LAB involved in milk fermentation has been comprehensively studied [47]. Hence, the subsequent objective of this study was to determine the extracellular proteolytic activity profile of LAB isolated from fermented vegetables. The results of the extracellular proteolytic activity of LAB isolates cultivated in MRS medium are shown in Figure 3A–H. The proteolytic activity profile was assessed under three different pH conditions. From the results obtained, the extracellular proteolytic activity of all the LAB isolates increased over 24 h of incubation, except for *P. acidilactici* UL-3. *P. acidilactici* UP-1 demonstrated the highest proteolytic activity under three different pH conditions in comparison to all tested LAB. The proteolytic activity increased gradually from 0 h and started to decline at 20 h of incubation. A comparatively high proteolytic activity at pH 5 (13.04 U/mg) was recorded by *P. pentosaceus* UP-2. The high AA production by *P. pentosaceus* UP-2 might be due to the presence of extracellular proteolytic enzymes, which was active in acidic conditions. The acidity of LAB growth medium caused by the production of organic acid might favor the proteolytic activity of certain LAB, for instance, *P. pentosaceus* UP-2 in this study [48,49,50]. Although the proteolytic activity of *L. plantarum* I-UL4 increased gradually from 0 h to 24 h of incubation, the AA production capacity was relatively low. The low AA production and high proteolytic activity of *L. plantarum* I-UL4 might be due to a limited biosynthesis capacity. Klaenhammer et al. [51] suggested that *L. plantarum* possessed a limited AA biosynthesis capacity, and it is compensated by a versatile proteolytic system.

On the other hand, *P. pentosaceus* UB-8 showed a major increase in proteolytic activity (5.15 U/mg) at pH 6.5 between 6 to 12 h of incubation. However, the proteolytic activity decreased rapidly after 12 h of incubation when the producer cells reached the stationary phase, as shown in Figure 2D. This was in line with the findings reported by Nissen-Meyer and Sletten [52], where the level of free extracellular proteases was the highest at the late exponential phase and early stationary phase of the producer cell. In acidic conditions, the proteolytic activity started to increase at 12 h to 14 h of incubation and maintained relatively high levels until 20 h. The result was in line with the AA production profile demonstrated by this LAB isolate, whereby most of the AAs reached the highest concentration between 14 to 20 h of incubation. This finding further suggested that the proteolytic activity of *P. pentosaceus* UB-8 might be involved in the production of AAs. Similar results were reported by Flambard et al. [53], in which the release of AAs from bovine milk casein by *L. lactis* was dependent on different types of proteinases. However, contradictory results were observed in *P. acidilactici* UL-3. The proteolytic activity of *P. acidilactici* UL-3 decreased with incubation time under three different pH conditions. However, an increasing production of AAs by *P. acidilactici* UL-3 was detected. This implied that there might be mechanisms other than biodegradation were responsible for the production of AAs by *P. acidilactici* UL-3.

Other LAB isolates (*P. acidilactici* UB-6, *P. pentosaceus* UL-2, and *P. pentosaceus* UL-6) showed relatively low proteolytic activities with no significant increases (*p* > 0.05) compared to *P. pentosaceus* UP-2. However, *P. acidilactici* UB-6 and *P. pentosaceus* UL-2 had the capability of producing 11 different AAs in relatively high amounts. *P. pentosaceus* UL-6 was the best proline producer, indicating that the proteolytic activity was not correlated proportionally to the production of AAs. However, to a certain extent, the proteolytic activity may contribute to the production of AAs. The increased proteolytic activity of *P. pentosaceus* UP-2 might be responsible for the increase of lysine concentration in the growth medium. Moreover, the increase in proteolytic activity of *P. pentosaceus* UB-8 between 6 to 14 h of incubation was in line with the increased levels of most AAs. Therefore, it can be concluded that the production of AAs by the studied LAB isolates was strain specific, whereby some of the LAB might produce AAs through a biodegradation pathway (proteolytic cleavage) or via other mechanisms involved in certain LAB isolates.

## 3. Materials and Methods

### 3.1. Bacterial Growth Conditions

A total of eight LAB isolates (*P. pentosaceus* UP-2, UL-2, UL-6, UB-8; *P. acidilactici* UP-1, UL-3, UB-6, and *Lactobacillus plantarum* I-UL4) isolated from Malaysian fermented food, *tapai ubi* [54], were obtained from the Laboratory of Industrial Biotechnology, Department of Bioprocess Technology, Universiti Putra Malaysia. The stock culture was maintained in de Man, Rogosa, and Sharpe (MRS) medium (Merck, Darmstadt, Germany) supplemented with 20% (*v/v*) glycerol (Merck, Darmstadt, Germany) and kept at −20 °C. The stock culture was revived by inoculating 1% (*v/v*) of the culture into 10 mL MRS and incubated at 30 °C anaerobically for 48 h. Then, 1% (*v/v*) of the 48 h culture was transferred into another 10 mL MRS broth. The broth was incubated at 30 °C for 24 h. After 24 h, the culture was streaked on MRS agar (Merck, Darmstadt, Germany) and incubated for 2 days at 30 °C. A single colony was picked and inoculated into 10 mL MRS broth for 48 h. A volume of 1% (*v/v*) of the 48 h culture was transferred into 10 mL MRS broth and incubated at 30 °C for 24 h [55].

The overall experimental protocols performed in this study are summarised in Figure 4. The extracellular proteolytic activity of the LAB isolates were determined prior to the productions of AAs.

### 3.2. Effect of pH on the Extracellular Proteolytic Activity of LAB

For the screening of proteolytic activity, 24 h LAB culture was adjusted to 10^9^ CFU/mL. A volume of 1% (*v/v*) culture was then inoculated into MRS medium and incubated at 30 °C for 10 h. The supernatant was collected by centrifugation at 10,000× *g* for 15 min and filtered through 0.2 µm cellulose acetate membrane (Sartorius, Göttingen, Germany) to obtain the cell-free supernatant (CFS). The CFS was used for the determination of proteolytic activity at different pH conditions by spectrophotometric methods, whereby azocasein (Sigma Aldrich, St. Louis, MO, USA) was used as a substrate [49].

### 3.3. Determination of Proteolytic Activity

Extracellular proteolytic activity was determined by using azocasein assay under three different pH conditions provided by 0.1 M sodium acetate (pH 5) (Merck, Darmstadt, Germany), 0.1 M sodium phosphate (pH 6.5) (Merck, Darmstadt, Germany), and 0.1 M Tris-HCl (pH 8) (Merck, Darmstadt, Germany) with minor modification, where the assay volume was reduced to half [49]. A volume of 250 µL of CFS was added to 500 µL of respective buffer solution containing 0.5% (*w/v*) sulphanilamide azocasein (Sigma Aldrich, St. Louis, MO, USA) and incubated at 37 °C for 30 min. The reaction was terminated by adding 750 µL of 10% (*w/v*) trichloroacetic acid (TCA) (Merck, Darmstadt, Germany) and incubated at room temperature for 30 min, followed by centrifugation at 12,000× *g* for 10 min to remove the precipitate. A volume of 600 µL of supernatant was mixed with 600 µL of 1 M NaOH (Merck, Darmstadt, Germany) and incubated at room temperature for 15 min. Then, the absorbance of the mixture was determined at 450 nm. The substrate blank was prepared by replacing the substrate with buffer, and the enzyme blank was prepared by replacing the enzyme with buffer. One unit (U/mg) of specific proteolytic activity was defined as the enzyme capability of hydrolyzing azocasein to produce a 0.001 change in absorbance per minute per amount of protein (mg) under the assay condition.

### 3.4. Determination of Solubilised Protein

Solubilized protein content of the CFS was determined by using the Bradford method [56] with bovine serum albumin (Sigma Aldrich, St. Louis, MO, USA) as the reference. A volume of 0.5 mL of appropriately diluted enzyme was added with 0.5 mL of Bradford reagent (Sigma Aldrich, St. Louis, MO, USA) and incubated at 4 °C for 5 min. The absorbance was measured at 595 nm.

### 3.5. Determination of the Amino Acid Production Profile of LAB

A volume of 10% (v/v) active LAB culture (adjusted to 10^9^ CFU/mL) was inoculated into MRS medium and incubated at 30 °C for 24 h. Samples were collected at 2 h intervals for the determination of cell population, residual reducing sugar concentration, AA production, and extracellular proteolytic activity.

#### 3.5.1. Determination of Cell Population

The cell population of LAB was determined by measuring the optical density at 600 nm (OD_600 nm_) by using a Varian Cary 50 spectrophotometer (Agilent Technologies, Santa Clara, CA, USA).

#### 3.5.2. Determination of Residual Reducing Sugar

The concentration of reducing sugar in CFS was determined with the dinitrosalicylic acid reagent (DNS) method [57]. A volume of 1 mL of appropriately diluted sample was added with 1 mL of DNS reagent. Then, 10 µL of 1 M NaOH was added. The mixture was boiled for 5 min before cooling down under running water. A volume of 10 mL of distilled water was added to the mixture and incubated for 20 min at room temperature. The absorbance of the mixture was measured at 540 nm. Glucose with different concentrations was used as reference.

#### 3.5.3. Determination of Amino Acids

The AA profile of CFS was determined by using an Agilent 1100 high pressure liquid chromatograph (Agilent Technologies, Santa Clara, CA, USA) equipped with a Zorbax Eclipse Plus C18 (4.6 × 150 mm, 3.5 µm) reverse phase column (Agilent Technologies, Santa Clara, CA, USA). Different concentrations of standard AA (Sigma Aldrich, St. Louis, MO, USA) were used as a reference. Sarcosine and norvaline (Sigma Aldrich, St. Louis, MO, USA) were used as internal standards. The AAs were derivatized by o-phthalaldehyde (OPA) and 9-fluorenylmethyl chloroformate (FMOC) (Merck, Darmstadt, Germany). The bound AAs were eluted with 40 mM of sodium dihydrogen phosphate monohydrate (Merck, Darmstadt, Germany) and a mixture of methanol:acetonitrile:deionized water (9:9:2) at a flow rate of 2 mL/min. The eluted AAs were detected at 338 nm and 262 nm absorbance, respectively. The results were analyzed by Agilent OpenLAB CDS ChemStation Edition A.02.05.021 software.

### 3.6. Statistical Analysis

All experiments were performed in three replicates; results were presented as mean ± standard error of mean analyzed with analysis of variance (ANOVA) using SAS 9.1.3 software (Cary, NC, USA).

## 4. Conclusions

All the tested LAB isolates possessed versatile extracellular proteolytic activities that were active over a broad pH range. Generally, the studied LAB isolates were able to produce isoleucine, proline, glutamate, and glycine, with *P. pentosaceus* UL-6 demonstrating the highest proline production ability. *P. pentosaceus* UP-2 was the best AA producer, as it had the capability to produce an array of AAs with the highest concentrations. The current findings showed that all the eight LAB isolates produced an array of different levels of AAs, indicating that the AA production was strain-dependent, whereby different LAB isolates exhibited different preferences and efficiencies in AA production. This warrants future research interest in investigating the relationship between proteolytic activity and AA production by LAB isolates. It could be useful and vital information to obtain if the extracellular proteolytic enzymes of LAB can be separated and characterized further (e.g., effects of inhibitors, substrate specificity and pH, etc.) to correlate with the specific AA production by each LAB.

## Figures and Tables

**Figure 1 ijms-20-01777-f001:**
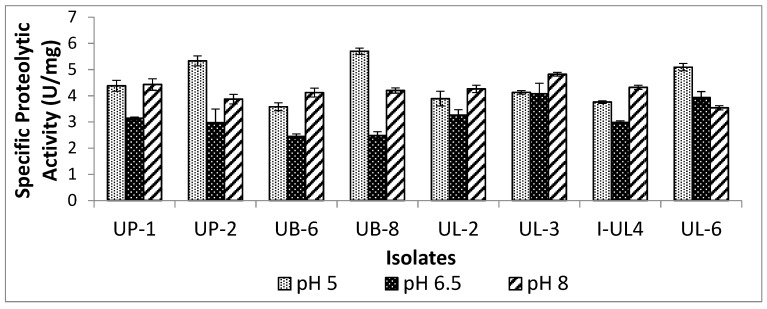
The extracellular proteolytic activity of lactic acid bacteria (LAB) under three different pH conditions.

**Figure 2 ijms-20-01777-f002:**
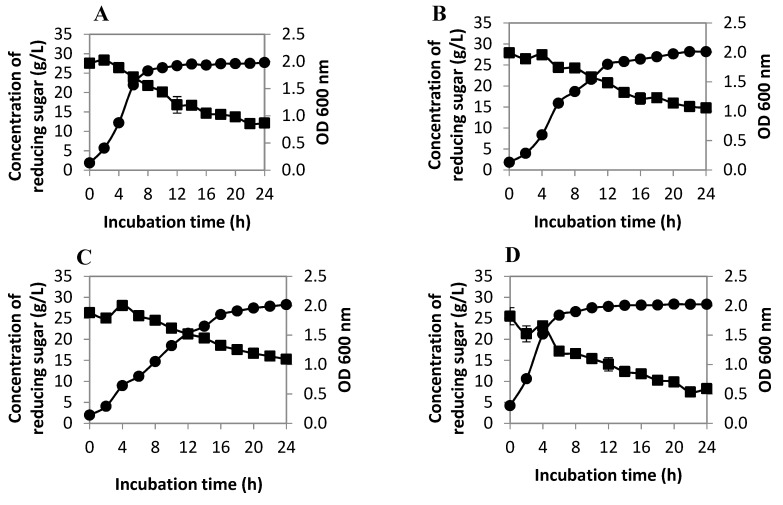
The cell growth profile (●) and reducing sugar consumption (■) of different LAB cultivated in de Man, Rogosa, and Sharpe (MRS) medium: (**A**) *Pediococcus acidilactici* UP-1; (**B**) *Pediococcus pentosaceus* UP-2; (**C**) *P. pentosaceus* UB-6; (**D**) *P. pentosaceus* UB-8; (**E**) *P. pentosaceus* UL-2; (**F)**
*P. acidilactici* UL-3; (**G**) *Lactobacillus plantarum* UL-4; and (**H**) *P. pentosaceus* UL-6.

**Figure 3 ijms-20-01777-f003:**
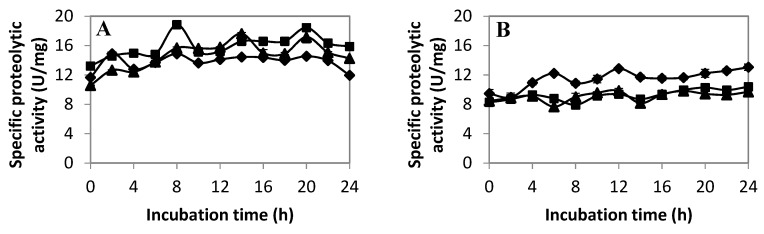
The specific extracellular proteolytic activities of different LAB isolates under pH5 (◆), pH 6.5 (■) and pH 8 (▲) conditions: (**A**) *P. acidilactici* UP-1; (**B)**
*P. pentosaceus* UP-2; (**C)**
*P. pentosaceus* UB-6; (**D**) *P. pentosaceus* UB-8; (**E**) *P. pentosaceus* UL-2; (**F**) *P. acidilactici* UL-3; (**G)**
*L. plantarum* UL-4; and (**H**) *P. pentosaceus* UL-6.

**Figure 4 ijms-20-01777-f004:**
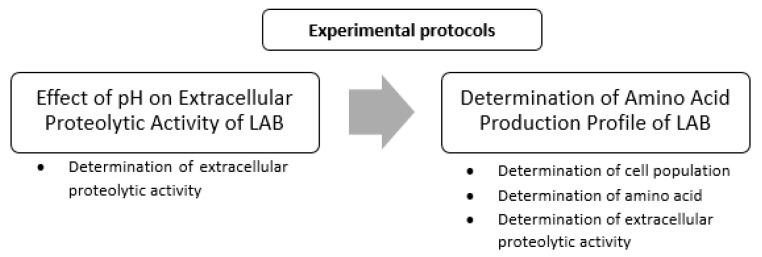
Overall experimental protocols.

**Table 1 ijms-20-01777-t001:** The amino acid production profile of *P. acidilactici* UP-1.

AA (mg/L)	Incubation Time (h)													Maximum Increment	
	0	2	4	6	8	10	12	14	16	18	20	22	24	Amount (mg/L)	Time (h)
**Asp**	44.49 ± 1.23 ^b^	44.59 ± 2.82 ^b^	48.39 ± 3.13 ^ab^	48.33 ± 1.64 ^ab^	52.18 ± 3.97 ^ab^	55.59 ± 1.66 ^a^	50.25 ± 2.55 ^ab^	50.06 ± 2.28 ^ab^	46.85 ± 1.16 ^b^	48.5 ± 0.71 ^ab^	46.1 ± 1.00 ^b^	49.3 ± 1.78 ^ab^	46.3 ± 3.10 ^b^	11.10	10
**Asn**	53.05 ± 2.10 ^g^	55.74 ± 1.61 ^efg^	54.57 ± 1.04 ^fg^	53.71 ± 0.67 ^g^	58.23 ± 1.50 ^defg^	65.98 ± 2.75 ^bcde^	60.54 ± 5.57 ^cdefg^	65.24 ± 2.14 ^bcdef^	74.83 ± 3.45 ^ab^	83.40 ± 6.42 ^a^	70.47 ± 3.56 ^bc^	67.03 ± 3.39 ^bcd^	62.52 ± 2.76 ^cdefg^	30.35	18
**Lys**	340.66 ± 9.12 ^a^	342.35 ± 5.89 ^a^	345.42 ± 7.92 ^a^	350.57 ± 8.65 ^a^	356.39 ± 9.00 ^a^	348.73 ± 9.91 ^a^	350.65 ± 3.68 ^a^	354.88 ± 9.22 ^a^	347.39 ± 5.85 ^a^	348.68 ± 3.76 ^a^	340.83 ± 8.81 ^a^	342.10 ± 1.96 ^a^	338.27 ± 4.32 ^a^	15.73	8
**Met**	93.67 ± 0.31 ^c^	99.36 ± 1.67 ^bc^	108.92 ± 6.91 ^abc^	110.77 ± 1.92 ^abc^	122.54 ± 8.02 ^a^	120.21 ± 6.95 ^ab^	118.05 ± 7.17 ^ab^	117.57 ± 7.45 ^ab^	114.51 ± 5.02 ^ab^	118.08 ± 6.34 ^ab^	115.52 ± 7.15 ^ab^	112.99 ± 8.80 ^abc^	113.68 ± 6.77 ^abc^	28.87	8
**Thr**	62.42 ± 4.41 ^cd^	65.36 ± 5.01 ^cd^	63.94 ± 4.67 ^cd^	57.79 ± 6.13 ^d^	84.38 ± 4.16 ^ab^	87.57 ± 3.53 ^a^	75.76 ± 8.23 ^abc^	74.50 ± 1.37 ^abc^	70.54 ± 1.29 ^bcd^	63.30 ± 6.96 ^cd^	64.74 ± 1.83 ^cd^	66.38 ± 3.05 ^cd^	72.30 ± 3.02 ^bcd^	25.15	10
**Ile**	83.71 ± 5.85 ^d^	102.14 ± 3.36 ^c^	103.96 ± 1.47 ^c^	114.18 ± 10.20 ^c^	130.18 ± 6.49 ^ab^	135.49 ± 5.19 ^a^	128.00 ± 4.26 ^b^	131.72 ± 4.71 ^ab^	129.88 ± 2.63 ^ab^	129.10 ± 4.07 ^ab^	124.81 ± 7.77 ^ab^	127.50 ± 2.97 ^b^	129.76 ± 4.25 ^ab^	51.78	10
**Glu**	222.01 ± 1.60 ^e^	220.81 ± 4.06 ^e^	249.61 ± 4.81 ^cd^	244.21 ± 5.22 ^d^	284.27 ± 8.25 ^ab^	290.55 ± 6.62 ^a^	273.72 ± 6.06 ^ab^	275.08 ± 7.96 ^ab^	267.30 ± 6.50 ^c^	280.88 ± 6.92 ^ab^	272.17 ± 6.08 ^ab^	281.05 ± 5.89 ^ab^	274.19 ± 8.15 ^ab^	68.54	10
**Pro**	40.50 ± 2.50 ^c^	45.84 ± 2.83 ^c^	47.58 ± 5.46 ^c^	45.77 ± 0.92 ^c^	54.75 ± 1.40 ^b^	58.50 ± 1.12 ^b^	55.59 ± 1.60 ^b^	61.03 ± 2.53 ^ab^	58.19 ± 1.75 ^ab^	62.30 ± 0.50 ^ab^	64.70 ± 1.06 ^a^	63.76 ± 2.54 ^a^	61.36 ± 3.12 ^ab^	24.20	20
**Arg**	238.25 ± 0.78 ^a^	223.86 ± 8.48 ^b^	196.86 ± 7.88 ^c^	174.03 ± 5.87 ^d^	170.99 ± 4.54 ^d^	151.10 ± 5.13 ^e^	84.79 ± 1.61 ^f^	57.02 ± 1.53 ^g^	0.00 ± 0.00 ^h^	0.00 ± 0.00 ^h^	0.00 ± 0.00 ^h^	0.00 ± 0.00 ^h^	0.00 ± 0.00 ^h^	0.00	0
**Ala**	180.54 ± 3.95 ^a^	181.39 ± 5.21 ^a^	184.62 ± 2.24 ^a^	183.95 ± 1.67 ^a^	194.11 ± 5.41 ^a^	193.95 ± 2.16 ^a^	192.19 ± 4.17 ^a^	194.12 ± 5.57 ^a^	189.12 ± 4.17 ^a^	186.65 ± 6.81 ^a^	186.87 ± 1.69 ^a^	185.51 ± 7.51 ^a^	184.51 ± 2.86 ^a^	13.58	14
**Val**	109.06 ± 8.31 ^a^	114.43 ± 9.13 ^a^	111.85 ± 3.76 ^a^	118.48 ± 6.94 ^a^	126.91 ± 8.90 ^a^	130.32 ± 5.04 ^a^	126.23 ± 1.47 ^a^	127.77 ± 7.72 ^a^	123.26 ± 6.65 ^a^	121.45 ± 7.06 ^a^	122.75 ± 5.89 ^a^	121.93 ± 2.28 ^a^	118.53 ± 7.02 ^a^	21.26	10
**Leu**	356.91 ± 9.43 ^d^	365.94 ± 4.96 ^cd^	374.44 ± 4.87 ^bcd^	379.66 ± 22.01 ^abcd^	381.71 ± 6.35 ^abcd^	393.85 ± 8.08 ^abc^	385.36 ± 4.21 ^abcd^	405.80 ± 9.19 ^a^	402.49 ± 1.61 ^ab^	407.03 ± 8.22 ^a^	390.77 ± 4.27 ^abc^	399.67 ± 6.93 ^ab^	384.66 ± 7.50 ^abcd^	50.12	18
**Phe**	152.11 ± 3.68 ^ab^	153.85 ± 3.11 ^a^	149.42 ± 2.95 ^ab^	135.77 ± 4.37 ^bc^	141.17 ± 8.82 ^abc^	132.20 ± 4.30 ^c^	132.06 ± 4.63 ^c^	127.67 ± 7.07 ^cd^	114.61 ± 5.77 ^de^	128.20 ± 6.25 ^cd^	124.42 ± 3.51 ^cd^	106.77 ± 5.66 ^ef^	93.86 ± 5.22 ^f^	1.74	2
**Trp**	88.33 ± 1.24 ^bc^	86.62 ± 3.03 ^bc^	84.67 ± 0.96 ^bc^	83.01 ± 7.91 ^c^	66.46 ± 0.61 ^a^	60.93 ± 1.94 ^b^	0.00 ± 0.00 ^d^	0.00 ± 0.00 ^d^	0.00 ± 0.00 ^d^	0.00 ± 0.00 ^d^	0.00 ± 0.00 ^d^	0.00 ± 0.00 ^d^	0.00 ± 0.00 ^d^	0.00	0
**Tyr**	80.43 ± 3.82 ^a^	78.98 ± 2.91 ^ab^	71.58 ± 1.38 ^abc^	69.85 ± 4.05 ^bc^	68.01 ± 5.02 ^c^	71.38 ± 3.41 ^abc^	66.81 ± 5.90 ^c^	41.07 ± 20.55 ^c^	22.5 ± 22.5 ^c^	0.00 ± 0.00 ^d^	0.00 ± 0.00 ^d^	0.00 ± 0.00 ^d^	0.00 ± 0.00 ^d^	0.00	0
**Ser**	116.21 ± 2.75 ^c^	119.57 ± 3.31 ^bc^	132.30 ± 2.23 ^ab^	131.71 ± 4.58 ^ab^	138.06 ± 5.20 ^a^	143.80 ± 3.87 ^a^	121.63 ± 2.03 ^bc^	108.90 ± 3.09 ^c^	84.25 ± 6.74 ^d^	61.96 ± 11.84 ^e^	0.00 ± 0.00 ^f^	0.00 ± 0.00 ^f^	0.00 ± 0.00 ^f^	27.59	10
**Gly**	75.04 ± 0.68 ^c^	73.66 ± 0.30 ^c^	74.75 ± 1.51 ^c^	72.24 ± 0.97 ^c^	97.02 ± 2.50 ^b^	112.44 ± 6.51 ^a^	104.06 ±5.14 ^ab^	101.75 ±2.85 ^ab^	102.01 ±7.35 ^ab^	106.13 ±2.65 ^ab^	101.51 ±0.80 ^ab^	107.85 ±5.75 ^ab^	105.31 ±4.56 ^ab^	37.39	10
**Cy2**	122.61 ± 6.07 ^d^	138.17 ± 3.76 ^cd^	149.74 ± 1.65 ^bc^	145.44 ± 5.20 ^bc^	151.19 ± 2.31 ^bc^	159.70 ± 0.97 ^ab^	161.92 ± 9.47 ^ab^	158.38 ± 2.23 ^ab^	171.85 ± 7.38 ^a^	160.82 ± 4.80 ^ab^	159.92 ± 2.47 ^ab^	152.45 ± 7.89 ^bc^	148.60 ± 6.72 ^bc^	49.24	16

Note: Glutamine and histidine were not detected. Asp, aspartate; Asn, asparagine; Lys, lysine; Met, methionine; Ile, isoleucine; Glu, glutamate; Pro, proline; Arg, arginine; Ala, alanine; Val, valine; Leu, leucine; Phe, phenylalanine; Trp, tryptophan; Tyr, tyrosine; Ser, serine; Gly, glycine; and Cy2, cystine. Values are means ± standard error of the mean (SEM), *n* = 3. Mean ± SEM within the same row sharing a common superscript letter are not significantly different at *p* > 0.05.

**Table 2 ijms-20-01777-t002:** The amino acid production profile of *P. pentosaceus* UP-2.

AA (mg/L)	Incubation Time (h)													Maximum Increment	
	0	2	4	6	8	10	12	14	16	18	20	22	24	Amount (mg/L)	Time (h)
**Asp**	42.00 ± 1.23 ^g^	42.85 ± 2.12 ^g^	46.42 ± 0.88 ^g^	52.34 ± 1.09 ^ef^	55.74 ± 0.97 ^cdef^	57.13 ± 1.34 ^cde^	51.49 ± 1.08 ^f^	60.53 ± 4.13 ^bc^	64.76 ± 0.40 ^ab^	65.83 ± 1.34 ^a^	55.92 ± 0.51 ^cdef^	55.28 ± 0.15 ^def^	57.77 ± 1.37 ^cd^	23.83	18
**Asn**	52.15 ± 0.85 ^a^	50.59 ± 2.88 ^a^	49.78 ± 1.51 ^ab^	48.52 ± 1.05 ^ab^	49.86 ± 2.50 ^ab^	48.21 ± 0.33 ^ab^	44.57 ± 0.35 ^b^	47.82 ± 4.24 ^b^	0.00 ± 0.00 ^c^	0.00 ± 0.00 ^c^	0.00 ± 0.00 ^c^	0.00 ± 0.00 ^c^	0.00 ± 0.00 ^c^	0	0
**Lys**	332.39± 19.03 ^bcd^	311.34 ± 4.88 ^d^	334.14 ± 25.56 ^bcd^	324.75 ± 7.38 ^bcd^	320.14 ± 0.36 ^cd^	306.72 ± 10.51 ^d^	313.18 ± 5.99 ^d^	335.32 ± 29.39 ^abcd^	364.37 ± 7.32 ^bc^	370.84 ± 13.51 ^ab^	372.07 ± 0.26 ^ab^	362.82 ± 21.34 ^abc^	383.18 ± 0.81 ^a^	50.79	24
**Met**	104.20 ± 5.00 ^f^	105.57 ± 1.61 ^f^	119.00 ± 4.79 ^e^	129.71 ± 0.33 ^d^	132.44 ± 2.04 ^d^	133.46 ± 3.76 ^d^	137.64 ± 1.78 ^cd^	139.55 ± 6.60 ^cd^	156.69 ± 1.29 ^a^	155.67 ± 1.87 ^a^	145.71 ± 0.87 ^bc^	138.99 ± 2.89 ^cd^	151.14 ± 0.92 ^a^	52.49	16
**Thr**	71.38 ± 2.08 ^h^	71.77 ± 2.23 ^h^	81.26 ± 4.11 ^gh^	87.87 ± 1.67 ^fg^	100.58 ± 0.21 ^de^	105.68 ± 0.30 ^cd^	94.94 ± 6.87 ^ef^	119.81 ± 8.65 ^ab^	127.49 ± 0.70 ^a^	130.16 ± 1.52 ^a^	120.28 ± 1.89 ^ab^	113.51 ± 0.44 ^bc^	124.87 ± 0.71 ^a^	58.78	18
**Ile**	100.20 ± 6.31 ^d^	104.15 ± 2.72 ^d^	113.03 ± 7.91 ^c^	139.29 ± 4.65 ^b^	162.10 ± 0.35 ^a^	139.19 ± 3.24 ^b^	130.53 ± 1.65 ^b^	133.08 ± 8.43 ^b^	135.59 ± 0.24 ^b^	136.64 ± 2.02 ^b^	132.20 ± 2.04 ^b^	126.35 ± 0.99 ^b^	134.85 ± 0.58 ^b^	61.9	8
**Glu**	233.74 ± 3.42 ^f^	233.84 ± 7.56 ^f^	252.29 ± 13.01 ^ef^	262.24 ± 5.60 ^de^	284.71 ± 1.84 ^cd^	285.51 ± 3.11 ^cd^	272.98 ± 3.70 ^de^	307.21 ± 21.76 ^abc^	322.33 ± 0.47 ^ab^	331.74 ± 2.57 ^a^	304.21 ± 2.44 ^bc^	305.37 ± 0.69 ^bc^	320.41 ± 4.83 ^ab^	98	18
**Pro**	45.72 ± 2.29 ^d^	45.85 ± 3.00 ^d^	46.07 ± 2.77 ^d^	55.35 ± 0.88 ^c^	61.99 ± 0.54 ^bc^	66.90 ± 2.33 ^ab^	68.00 ± 0.88 ^ab^	65.40 ± 1.13 ^ab^	72.89 ± 1.47 ^a^	73.46 ± 2.49 ^a^	73.57 ± 5.10 ^a^	73.69 ± 4.51 ^a^	70.74 ± 0.98 ^a^	27.97	22
**Arg**	184.04 ± 4.80 ^ab^	186.04 ± 13.80 ^ab^	195.74 ± 0.53 ^a^	193.36 ± 2.92 ^a^	193.93 ± 0.96 ^a^	171.91 ± 1.64 ^b^	135.41 ± 4.12 ^c^	124.19 ± 6.78 ^c^	0.00 ± 0.00 ^d^	0.00 ± 0.00 ^d^	0.00 ± 0.00 ^d^	0.00 ± 0.00 ^d^	0.00 ± 0.00 ^d^	11.7	4
**Ala**	163.10 ± 6.42 ^h^	165.94 ± 10.87 ^gh^	178.11 ± 1.72 ^fgh^	183.03 ± 3.23 ^efg^	192.69 ± 0.44 ^cdef^	189.74 ± 0.19 ^cdef^	185.52 ± 2.86 ^def^	206.14 ± 16.66 ^bc^	216.50 ± 0.65 ^ab^	226.89 ± 1.54 ^a^	219.55 ± 1.97 ^ab^	205.44 ± 5.74 ^bcd^	199.98 ± 1.80 ^bcde^	63.79	18
**Val**	87.28 ± 2.24 ^f^	83.40 ± 1.82 ^f^	82.60 ± 5.53 ^f^	82.92 ± 0.40 ^f^	86.33 ± 1.47 ^f^	89.36 ± 2.54 ^ef^	89.25 ± 1.14 ^ef^	95.94 ± 4.00 ^de^	103.79 ± 0.35 ^bc^	108.39 ± 0.82 ^ab^	104.36 ± 1.77 ^bc^	99.77 ± 1.24 ^cd^	112.89 ± 1.57 ^a^	25.61	24
**Leu**	357.86 ± 9.06 ^c^	359.20 ± 7.46 ^c^	382.01 ± 21.38 ^bc^	363.62 ± 11.63 ^c^	387.07 ± 2.32 ^bc^	377.13 ± 2.02 ^bc^	379.77 ± 13.43 ^bc^	411.17 ± 32.76 ^ab^	437.75 ± 0.45 ^a^	453.28 ± 4.61 ^a^	434.79 ± 6.94 ^a^	428.33 ± 10.83 ^a^	452.54 ± 2.82 ^a^	95.42	18
**Phe**	141.52 ± 3.06 ^a^	133.00 ± 4.80 ^ab^	128.18 ± 7.82 ^b^	113.66 ± 0.34 ^c^	112.54 ± 0.24 ^c^	109.08 ± 2.00 ^c^	107.55 ± 1.53 ^c^	110.14 ± 7.10 ^c^	109.32 ± 0.25 ^c^	110.67 ± 1.43 ^c^	110.56 ± 3.09 ^c^	103.07 ± 3.72 ^c^	114.04 ± 2.69 ^c^	0	0
**Trp**	73.91 ± 2.13 ^b^	81.64 ± 4.71 ^a^	82.25 ± 3.84 ^a^	70.46 ± 0.59 ^b^	0.00 ± 0.00 ^c^	0.00 ± 0.00 ^c^	0.00 ± 0.00 ^c^	0.00 ± 0.00 ^c^	0.00 ± 0.00 ^c^	0.00 ± 0.00 ^c^	0.00 ± 0.00 ^c^	0.00 ± 0.00 ^c^	0.00 ± 0.00 ^c^	8.34	4
**Tyr**	66.93 ± 2.60 ^b^	74.87 ± 6.90 ^a^	75.52 ± 5.58 ^a^	60.35 ± 0.92 ^b^	0.00 ± 0.00 ^c^	0.00 ± 0.00 ^c^	0.00 ± 0.00 ^c^	0.00 ± 0.00 ^c^	0.00 ± 0.00 ^c^	0.00 ± 0.00 ^c^	0.00 ± 0.00 ^c^	0.00 ± 0.00 ^c^	0.00 ± 0.00 ^c^	8.59	4
**Ser**	116.85 ± 4.63 ^d^	118.2 ± 2.5 ^d^	130.6 ± 8.3 ^cd^	129.6 ± 2.6 ^cd^	138.5 ± 0.4 ^c^	137.17 ± 0.30 ^c^	137.91 ± 2.02 ^c^	153.93 ± 12.63 ^b^	162.60 ± 0.97 ^ab^	167.43 ± 1.71 ^ab^	160.08 ± 1.32 ^ab^	161.62 ± 3.78 ^ab^	171.12 ± 0.91 ^a^	54.27	24
**Gly**	61.15 ± 3.33 ^g^	63.02± 4.09 ^fg^	71.12± 1.71 ^def^	70.44± 1.10 ^efg^	81.02± 0.65 ^bc^	74.41± 1.10 ^cde^	69.79± 1.59 ^efg^	80.10± 7.97 ^bcd^	85.74 ±1.17 ^ab^	91.87 ± 0.61 ^a^	86.16 ±1.93 ^ab^	88.71 ±2.57 ^ab^	94.90 ± 1.81 ^a^	33.75	24
**Cy2**	91.72 ± 13.57 ^f^	107.43 ± 10.22 ^def^	134.01 ± 1.88 ^bcd^	119.77 ± 4.13 ^cdef^	146.32 ± 6.46 ^abc^	125.54 ± 10.11 ^bcde^	95.90 ± 4.17 ^ef^	154.36 ± 20.80 ^ab^	167.60 ± 10.06 ^a^	173.72 ± 8.97 ^a^	113.75 ± 7.96 ^def^	116.63 ± 0.48 ^cdef^	126.70 ± 4.76 ^bcde^	82	18

Note: Glutamine and histidine were not detected. Asp, aspartate; Asn, asparagine; Lys, lysine; Met, methionine; Ile, isoleucine; Glu, glutamate; Pro, proline; Arg, arginine; Ala, alanine; Val, valine; Leu, leucine; Phe, phenylalanine; Trp, tryptophan; Tyr, tyrosine; Ser, serine; Gly, glycine; Cy2, cystine. Values are means ± standard error of the mean (SEM), n = 3. Mean ± SEM within the same row sharing a common superscript letter are not significantly different at P > 0.05.

**Table 3 ijms-20-01777-t003:** The amino acid production profile of *P. acidilactici* UB6.

AA (mg/L)	Incubation Time (h)													Maximum Increment	
	0	2	4	6	8	10	12	14	16	18	20	22	24	Amount (mg/L)	Time (h)
**Asp**	36.47 ± 4.50 ^a^	33.89 ± 0.67 ^abc^	29.64 ± 1.55 ^bcd^	36.43 ± 2.22 ^a^	35.22 ± 0.80 ^ab^	36.82 ± 1.20 ^a^	31.55 ± 1.71 ^abcd^	27.34 ± 0.47 ^d^	31.89 ± 0.26 ^abcd^	31.66± 1.03 ^abcd^	30.39 ± 1.97 ^bcd^	28.77 ± 0.14 ^cd^	29.18 ± 0.93 ^cd^	0.36	10
**Asn**	48.56 ± 2.41 ^a^	42.92 ± 1.69 ^b^	38.66 ± 3.77 ^bc^	37.40 ± 1.89 ^bc^	33.75 ± 1.83 ^c^	34.49 ± 2.05 ^c^	36.54 ± 1.76 ^c^	34.26 ± 0.09 ^c^	38.14 ± 0.43 ^bc^	34.79 ± 0.62 ^cd^	36.50 ± 0.27 ^cd^	34.52 ± 1.65 ^cd^	33.48 ± 2.04 ^cd^	0	0
**Lys**	280.87 ± 2.82 ^bcd^	257.85 ± 9.62 ^ef^	253.09 ± 1.11 ^f^	265.62 ± 4.45 ^cdef^	277.30 ± 3.34 ^bcde^	261.80 ± 13.39 ^def^	283.67 ± 0.28 ^bc^	271.29 ± 6.86 ^bcdef^	288.82 ± 2.28 ^b^	316.65 ± 5.85 ^a^	281.06 ± 4.69 ^bcd^	264.17 ± 5.72 ^cdef^	270.29 ± 5.03 ^bcdef^	35.78	18
**Met**	49.25 ± 0.06 ^d^	50.21 ± 0.10 ^cd^	54.21 ± 0.33 ^abc^	50.65 ± 3.23 ^cd^	53.08 ± 1.09 ^abcd^	52.04 ± 1.54 ^bcd^	54.29 ± 0.96 ^abc^	49.20 ± 0.99 ^d^	55.53 ± 0.47 ^ab^	56.42 ± 1.23 ^a^	53.07 ± 1.03 ^abcd^	53.24 ± 0.41 ^abcd^	52.21 ± 1.69 ^abcd^	7.18	18
**Thr**	56.83 ± 1.31 ^e^	55.64 ± 0.72 ^e^	59.77 ± 2.24 ^de^	69.04 ± 3.76 ^bc^	73.11 ± 5.27 ^abc^	66.38 ± 2.04 ^cd^	77.32 ± 2.52 ^a^	68.47 ± 0.65 ^c^	73.20 ± 0.19 ^abc^	76.71 ± 2.91 ^ab^	71.15 ± 1.64 ^abc^	77.16 ± 1.68 ^a^	76.63 ± 1.49 ^ab^	20.49	12
**Ile**	69.72 ± 3.00 ^c^	63.02 ± 1.76 ^d^	72.09 ± 1.53 ^bc^	77.74 ± 3.97 ^bc^	79.35 ± 2.63 ^a^	79.04 ± 0.79 ^a^	78.16 ± 0.79 ^a^	77.48 ± 0.29 ^bc^	83.60 ± 0.50 ^a^	81.57 ± 1.36 ^a^	81.51 ± 1.94 ^a^	78.36 ± 0.96 ^a^	77.93 ± 0.11 ^bc^	13.89	16
**Glu**	176.10 ± 11.28 ^cd^	167.47 ± 6.77 ^d^	181.38 ± 3.84 ^bcd^	206.26 ± 11.47 ^a^	194.23 ± 8.70 ^abc^	197.26 ± 4.79 ^ab^	197.14 ± 7.19 ^ab^	198.31 ± 0.68 ^ab^	202.49 ± 2.16 ^a^	195.86 ± 4.11 ^abc^	204.19 ± 2.73 ^a^	187.21 ± 2.70 ^abc^	192.91 ± 0.46 ^abc^	30.16	6
**Pro**	53.26 ± 0.50 ^e^	48.23 ± 0.09 ^f^	56.73 ± 3.86 ^de^	61.93 ± 1.27 ^abc^	65.21 ± 1.98 ^abc^	60.86 ± 1.43 ^bcd^	60.63 ± 1.14 ^cd^	61.49 ± 0.60 ^abcd^	63.90 ± 2.22 ^abc^	65.05 ± 0.77 ^abc^	62.04 ± 0.42 ^abc^	65.97 ± 1.10 ^ab^	66.34 ± 1.03 ^a^	13.08	24
**Arg**	151.92 ± 3.36 ^a^	136.53 ± 2.06 ^b^	138.67 ± 2.95 ^b^	127.31 ± 0.46 ^c^	81.59 ± 0.31 ^d^	36.66 ± 0.95 ^e^	3.42 ± 0.69 ^h^	4.22 ± 0.84 ^hg^	9.95 ± 3.34 ^f^	9.78 ± 1.35 ^fg^	3.73 ± 1.01 ^h^	4.49 ± 0.39 ^afgh^	4.73 ± 0.11 ^fgh^	0.00	0
**Ala**	119.73 ± 5.02 ^cd^	113.20 ± 2.87 ^d^	120.41 ± 2.86 ^cd^	120.92 ± 5.47 ^cd^	119.26 ± 7.08 ^cd^	120.77 ± 3.11 ^cd^	122.41 ± 2.85 ^cd^	127.30 ± 0.07 ^bc^	136.05 ± 1.93 ^ab^	142.69 ± 2.66 ^a^	138.88 ± 2.74 ^a^	137.85 ± 2.03 ^ab^	140.46 ± 0.27 ^a^	22.95	18
**Val**	77.42 ± 3.34 ^cd^	73.84 ± 1.44 ^d^	81.86 ± 2.36 ^bc^	90.22 ± 4.51 ^a^	91.56 ± 5.38 ^a^	90.92 ± 1.21 ^a^	92.20 ± 0.14 ^a^	89.65 ± 0.33 ^ab^	92.95 ± 0.41 ^a^	93.76 ± 1.56 ^a^	92.94 ± 2.83 ^a^	87.95 ± 0.29 ^ab^	89.18 ± 0.74 ^ab^	16.34	18
**Leu**	242.48 ± 5.95 ^ab^	217.21 ± 2.48 ^c^	237.33 ± 9.48 ^abc^	230.35 ± 2.65 ^bc^	232.80 ± 11.22 ^abc^	230.00 ± 10.01 ^bc^	235.80 ± 3.80 ^abc^	232.29 ± 2.76 ^bc^	254.11 ± 6.25 ^a^	240.59 ± 4.17 ^ab^	243.55 ± 6.33 ^ab^	240.59 ± 5.86 ^ab^	241.41 ± 3.22 ^ab^	11.63	16
**Phe**	122.16 ± 2.28 ^a^	113.90 ± 0.42 ^b^	108.57 ± 2.97 ^bc^	111.85 ± 5.93 ^b^	109.23 ± 2.00 ^bc^	108.38 ± 3.27 ^bc^	101.67 ± 1.51 ^cd^	97.11 ± 0.97 ^de^	99.12 ± 0.33 ^ed^	92.79 ± 2.81 ^e^	92.57 ± 2.79 ^e^	79.46 ± 0.25 ^f^	81.96 ± 0.48 ^f^	0.00	0
**Trp**	54.90 ± 0.26 ^bc^	57.81 ± 1.49 ^ab^	49.89 ± 1.58 ^d^	49.96 ± 3.10 ^d^	59.65 ± 0.96 ^ab^	60.90 ± 1.38 ^a^	54.96 ± 1.85 ^bc^	50.87 ± 0.19 ^cd^	58.15 ± 2.17 ^ab^	60.00 ± 0.93 ^a^	51.63 ± 0.06 ^cd^	49.80 ± 0.42 ^d^	50.35 ± 1.93 ^cd^	6.00	10
**Tyr**	28.18 ± 0.05 ^a^	26.70 ± 0.51 ^a^	23.67 ± 1.06 ^a^	26.70 ± 0.76 ^a^	26.73 ± 0.08 ^a^	26.88 ± 0.23 ^a^	25.01 ± 0.11 ^a^	23.48 ± 0.96 ^a^	26.26 ± 1.13 ^a^	28.24 ± 1.69 ^a^	22.77 ± 0.12 ^a^	24.06 ± 1.13 ^a^	22.57 ± 0.13 ^a^	0.06	0
**Ser**	79.77 ± 2.31 ^e^	80.54 ± 0.92 ^e^	87.92 ± 0.20 ^cde^	86.63 ± 3.27 ^de^	91.20 ± 5.84 ^bcd^	90.98 ± 6.72 ^bcd^	93.75 ± 1.01 ^abcd^	96.30 ± 2.39 ^abcd^	102.43 ± 1.10 ^a^	100.03 ± 1.56 ^ab^	100.57 ± 2.91 ^ab^	97.01 ± 1.62 ^abc^	97.41 ± 3.04 ^abc^	22.66	16
**Gly**	44.40 ± 0.68 ^d^	44.15 ± 0.08 ^d^	50.98 ± 1.52 ^c^	54.04 ± 1.87 ^bc^	54.70 ± 4.05 ^bc^	53.93 ± 4.08 ^bc^	58.88 ± 0.87 ^ab^	57.45 ± 1.98 ^abc^	63.79 ± 1.04 ^a^	61.67 ± 0.95 ^a^	62.23 ± 2.59 ^a^	61.85 ± 0.99 ^a^	62.94 ± 0.56 ^a^	19.39	16
**Cy2**	106.99 ± 4.08 ^bcd^	95.74 ± 1.68 ^d^	99.30 ± 4.86 ^d^	98.22 ± 3.06 ^d^	98.06 ± 4.41 ^d^	100.73 ± 4.84 ^d^	103.80 ± 6.07 ^cd^	99.19 ± 7.21 ^d^	119.92 ±3.94 ^abc^	121.73 ± 1.49 ^ab^	120.03 ± 9.72 ^abc^	120.28 ± 1.94 ^abc^	133.13 ± 7.15 ^a^	26.14	24

Note: Glutamine and histidine were not detected. Asp, aspartate; Asn, asparagine; Lys, lysine; Met, methionine; Ile, isoleucine; Glu, glutamate; Pro, proline; Arg, arginine; Ala, alanine; Val, valine; Leu, leucine; Phe, phenylalanine; Trp, tryptophan; Tyr, tyrosine; Ser, serine; Gly, glycine; Cy2, cystine. Values are means ± standard error of the mean (SEM), *n* = 3. Mean ± SEM within the same row sharing a common superscript letter are not significantly different at *p* > 0.05.

**Table 4 ijms-20-01777-t004:** The amino acid production profile of *P. pentosaceus* UB-8.

AA (mg/L)	Incubation Time (h)													Maximum Increment	
	0	2	4	6	8	10	12	14	16	18	20	22	24	Amount (mg/L)	Time (h)
**Asp**	57.98 ± 1.18 ^a^	54.90 ± 2.67 ^ab^	56.22 ± 2.63 ^ab^	51.70 ± 2.47 ^b^	55.83 ± 3.37 ^ab^	57.29 ± 1.77 ^a^	56.46 ± 1.44 ^ab^	0.00 ± 0.00 ^c^	0.00 ± 0.00 ^c^	0.00 ± 0.00 ^c^	0.00 ± 0.00 ^c^	0.00 ± 0.00 ^c^	0.00 ± 0.00 ^c^	0.00	0
**Asn**	75.11 ± 0.81 ^a^	71.18 ± 4.77 ^ab^	72.34 ± 0.57 ^ab^	63.46 ± 3.29 ^c^	71.01 ± 4.55 ^ab^	71.64 ± 0.68 ^ab^	65.62 ± 1.06 ^bc^	46.97 ± 23.51 ^c^	0.00 ± 0.00 ^d^	0.00 ± 0.00 ^d^	0.00 ± 0.00 ^d^	0.00 ± 0.00 ^d^	0.00 ± 0.00 ^d^	0.00	0
**Lys**	440.20 ± 5.70 ^a^	451.33 ± 8.68 ^a^	436.07 ± 5.94 ^a^	401.44 ± 4.19 ^bc^	393.75 ± 6.83 ^c^	393.91 ± 1.37 ^c^	396.46 ± 3.22 ^c^	416.32 ± 5.07 ^b^	392.89 ± 6.21 ^c^	371.98 ± 5.54 ^d^	353.99 ± 7.18 ^d^	334.04 ± 8.66 ^e^	363.47 ± 8.77 ^d^	11.13	4
**Met**	100.81 ± 0.16 ^d^	100.91 ± 3.06 ^d^	113.87 ± 1.06 ^bcd^	106.57 ± 2.35 ^d^	109.80 ± 7.81 ^cd^	119.96 ± 4.28 ^abc^	119.0 ± 2.02 ^abc^	128.84 ± 0.07 ^a^	123.90 ± 1.32 ^ab^	125.77 ± 4.08 ^ab^	132.71 ± 7.93 ^a^	128.99 ± 5.37 ^a^	126.34 ± 4.55 ^ab^	31.90	20
**Thr**	92.89 ± 0.78 ^abc^	83.18 ± 5.30 ^c^	91.07 ± 6.15 ^abc^	88.30 ± 4.65 ^bc^	99.57 ± 6.62 ^abc^	107.96 ± 0.24 ^a^	100.49 ± 1.24 ^ab^	101.80 ± 2.22 ^ab^	100.00 ± 5.91 ^abc^	94.78 ± 8.43 ^abc^	68.25 ± 8.10 ^d^	54.26 ± 2.11 ^d^	55.36 ± 4.82 ^d^	15.06	10
**Ile**	123.70 ± 2.52 ^c^	146.71 ± 8.04 ^ab^	152.87 ± 3.66 ^a^	146.59 ± 7.59 ^ab^	152.09 ± 10.01 ^a^	155.07 ± 3.28 ^a^	148.25 ± 6.33 ^ab^	163.48 ± 4.09 ^a^	153.11 ± 1.19 ^a^	150.79 ± 6.04 ^ab^	127.52 ± 5.90 ^c^	122.54 ± 4.61 ^c^	133.48 ± 3.62 ^bc^	39.79	14
**Glu**	244.46 ± 5.57 ^e^	244.70 ± 6.49 ^e^	251.28 ± 8.53 ^de^	253.62 ± 4.49 ^de^	256.88 ± 3.49 ^cde^	280.46 ± 3.55 ^ab^	278.00 ± 2.47 ^abc^	290.27 ± 9.49 ^a^	271.35 ± 5.30 ^abcd^	263.94 ± 2.42 ^bcde^	264.36 ± 9.63 ^bcde^	262.52 ± 7.72 ^bcde^	262.39 ± 8.84 ^bcde^	45.81	14
**Pro**	37.24 ± 0.31 ^g^	37.90 ± 0.93 ^g^	46.25 ± 1.35 ^fg^	57.19 ± 1.83 ^ef^	65.22 ± 2.86 ^de^	71.56 ± 7.06 ^bcd^	80.46 ± 9.69 ^abc^	70.61 ± 4.57 ^cd^	75.65 ± 3.36 ^abcd^	79.55 ± 2.83 ^abc^	86.13 ± 2.49 ^a^	81.12 ± 5.04 ^ab^	85.06 ± 1.32 ^abc^	48.89	20
**Arg**	289.51 ± 2.16 ^a^	255.66 ± 9.01 ^b^	252.12 ± 6.78 ^b^	185.52 ± 8.83 ^c^	134.92 ± 1.87 ^d^	85.91 ± 4.61 ^e^	0.00 ± 0.00 ^f^	0.00 ± 0.00 ^f^	0.00 ± 0.00 ^f^	0.00 ± 0.00 ^f^	0.00 ± 0.00 ^f^	0.00 ± 0.00 ^f^	0.00 ± 0.00 ^f^	0.00	0
**Ala**	218.94 ± 3.05 ^a^	217.08 ± 8.85 ^a^	217.19 ± 3.09 ^a^	193.60 ± 7.47 ^b^	195.88 ± 5.18 ^b^	183.84 ± 6.23 ^bc^	171.54 ± 2.78 ^cd^	179.86 ± 0.73 ^bcd^	173.01 ± 2.24 ^cd^	171.58 ± 3.35 ^cd^	164.20 ± 7.56 ^d^	166.71 ± 4.38 ^cd^	174.27 ± 7.02 ^cd^	0.00	0
**Val**	99.46 ± 3.79 ^cd^	100.54 ± 7.85 ^cd^	107.27 ± 7.50 ^bc^	107.50 ± 6.48 ^bc^	116.49 ± 6.91 ^abc^	124.67 ± 1.88 ^ab^	121.6 3± 7.14 ^ab^	129.84 ± 6.27 ^a^	124.2 ± 2.79 ^abc^	118.7 ± 7.15 ^ab^	84.10 ± 5.29 ^de^	79.27 ± 1.34 ^e^	88.05 ± 3.17 ^de^	30.38	14
**Leu**	430.99 ± 1.80 ^cdef^	432.12 ± 2.26 ^cdef^	443.27 ± 4.87 ^bc^	434.76 ± 4.65 ^bcdef^	434.55 ± 8.63 ^bcdef^	441.52 ± 4.94 ^bcd^	449.68 ± 7.16 ^ab^	462.72 ± 3.23 ^a^	440.30 ± 7.47 ^bcd^	438.22 ± 2.34 ^cdef^	425.77 ± 0.90 ^def^	420.06 ± 3.30 ^f^	423.88 ± 4.78 ^ef^	31.73	14
**Phe**	145.38 ± 6.60 ^b^	128.71 ± 8.37 ^bcd^	137.25 ± 8.31 ^bc^	111.48 ± 8.21 ^d^	118.73 ± 7.45 ^cd^	11.8.84 ± 6.10 ^cd^	120.73 ± 5.88 ^cd^	122.46 ± 4.29 ^cd^	117.31 ± 3.36 ^cd^	115.50 ± 3.69 ^d^	128.33 ± 3.36 ^bcd^	167.96 ± 2.28 ^a^	183.25 ± 7.87 ^a^	37.87	24
**Trp**	124.92 ± 1.82 ^a^	115.80 ± 6.82 ^a^	120.21 ± 4.47 ^a^	109.97 ± 3.29 ^ab^	121.31 ± 10.91 ^a^	117.66 ± 5.71 ^a^	117.69 ± 4.14 ^a^	123.80 ± 2.09 ^a^	119.80 ± 2.11 ^a^	112.42 ± 4.65 ^ab^	100.15 ± 4.93 ^b^	0.00 ± 0.00 ^c^	0.00 ± 0.00 ^c^	0.00	0
**Tyr**	83.04 ± 1.57 ^a^	82.59 ± 4.87 ^a^	83.90 ± 3.31 ^a^	78.11 ± 3.61 ^a^	79.92 ± 5.90 ^a^	78.70 ± 3.61 ^a^	51.04 ± 25.53 ^a^	0.00 ± 0.00 ^b^	0.00 ± 0.00 ^b^	0.00 ± 0.00 ^b^	0.00 ± 0.00 ^b^	0.00 ± 0.00 ^b^	0.00 ± 0.00 ^b^	0.87	4
**Ser**	138.54 ± 2.09 ^ab^	134.52 ± 7.06 ^abc^	140.82 ± 1.51 ^a^	127.12 ± 8.08 ^ab^	120.16 ± 9.67 ^bc^	121.88 ± 0.83 ^bc^	115.87 ± 4.43 ^c^	119.26 ± 3.16 ^c^	117.68 ± 3.24 ^c^	119.35 ± 2.13 ^c^	127.32 ± 7.52 ^abc^	134.03 ± 5.55 ^abc^	132.45± 8.09 ^abc^	2.27	4
**Gly**	83.53 ± 1.85 ^d^	77.90 ± 6.94 ^d^	86.73 ± 3.26 ^cd^	80.31 ± 5.00 ^d^	100.67 ± 9.01 ^bc^	101.07 ± 3.93 ^bc^	100.36 ± 1.90 ^bc^	101.20 ± 3.00 ^bc^	97.73 ± 6.10 ^bc^	107.92 ± 3.04 ^b^	126.46 ± 0.82 ^a^	124.11 ± 2.94 ^a^	122.66 ± 4.18 ^a^	42.94	20
**Cy2**	176.31 ± 6.49 ^a^	172.49 ± 7.9 ^a^	173.99 ± 6.76 ^a^	172.53 ± 7.20 ^a^	173.27 ± 9.76 ^a^	171.17 ± 4.10 ^a^	174.64 ± 6.07 ^a^	173.67 ± 7.17 ^a^	176.47 ± 7.48 ^a^	178.63 ± 8.11 ^a^	190.63 ± 8.67 ^a^	185.97 ± 3.73 ^a^	178.97 ± 9.35 ^a^	14.32	20

Note: Glutamine and histidine were not detected. Asp, aspartate; Asn, asparagine; Lys, lysine; Met, methionine; Ile, isoleucine; Glu, glutamate; Pro, proline; Arg, arginine; Ala, alanine; Val, valine; Leu, leucine; Phe, phenylalanine; Trp, tryptophan; Tyr, tyrosine; Ser, serine; Gly, glycine; Cy2, cystine. Values are means ± standard error of the mean (SEM), *n* = 3. Mean ± SEM within the same row sharing a common superscript letter are not significantly different at *p* > 0.05.

**Table 5 ijms-20-01777-t005:** The amino acid production profile of *P. pentosaceus* UL-2.

AA (mg/L)	Incubation Time (h)													Maximum Increment	
	0	2	4	6	8	10	12	14	16	18	20	22	24	Amount (mg/L)	Time (h)
**Asp**	43.30 ± 1.48 ^a^	44.28 ± 2.44 ^a^	43.04 ± 0.64 ^a^	40.77 ± 3.58 ^a^	40.60 ± 1.84 ^a^	39.40 ± 1.44 ^a^	40.49 ± 1.15 ^a^	28.81 ± 14.41 ^a^	0.00 ± 0.00 ^b^	0.00 ± 0.00 ^b^	0.00 ± 0.00 ^b^	0.00 ± 0.00 ^b^	0.00 ± 0.00 ^b^	0.98	2
**Asn**	53.15 ± 2.24 ^a^	51.34 ± 2.53 ^a^	43.29 ± 1.29 ^b^	40.84 ± 4.11 ^b^	43.70 ± 2.84 ^b^	27.72 ± 13.86 ^b^	0.00 ± 0.00 ^c^	0.00 ± 0.00 ^c^	0.00 ± 0.00 ^c^	0.00 ± 0.00 ^c^	0.00 ± 0.00 ^c^	0.00 ± 0.00 ^c^	0.00 ± 0.00 ^c^	0.00	0
**Lys**	208.29 ± 5.04 ^a^	214.11 ± 6.76 ^a^	211.15 ± 6.87 ^a^	196.66 ± 4.33 ^a^	203.43 ± 8.84 ^a^	193.46 ± 7.98 ^a^	208.04 ± 9.82 ^a^	202.63 ± 1.18 ^a^	208.495 ± 9.31 ^a^	162.31 ± 9.21 ^b^	149.98 ± 8.49 ^bc^	141.98 ± 3.14 ^bc^	136.19 ± 8.04 ^c^	5.83	2
**Met**	48.46 ± 2.31 ^ab^	54.49 ± 3.54 ^a^	49.73 ± 1.97 ^ab^	54.65 ± 4.96 ^a^	58.38 ± 2.46 ^a^	54.12 ± 3.74 ^a^	53.38 ± 1.81 ^ab^	57.36 ± 1.98 ^a^	55.17 ± 2.24 ^a^	52.67 ± 6.63 ^ab^	50.90 ± 2.81 ^ab^	43.45 ± 1.76 ^b^	43.17 ± 1.02 ^b^	9.92	8
**Thr**	35.36 ± 4.39 ^d^	37.70 ± 5.61 ^cd^	43.99 ± 1.25 ^abcd^	53.92 ± 5.43 ^a^	53.73 ± 1.86 ^a^	52.73 ± 4.06 ^a^	50.39 ± 1.81 ^ab^	47.55 ± 2.51 ^abc^	54.34 ± 0.62 ^a^	47.91 ± 4.59 ^abc^	44.35 ± 0.99 ^abcd^	39.37 ± 0.23 ^cd^	40.14 ± 1.35 ^bcd^	18.98	16
**Ile**	59.05 ± 2.40 ^c^	66.67 ± 4.36 ^abc^	75.21 ± 6.56 ^ab^	78.43 ± 6.11 ^a^	74.38 ± 3.84 ^ab^	75.65 ± 6.39 ^a^	77.86 ± 4.71 ^a^	77.92 ± 2.94 ^a^	78.61 ± 4.48 ^a^	68.17 ± 7.85 ^abc^	60.18 ± 3.44 ^bc^	54.69 ± 0.16 ^c^	54.19 ± 1.40 ^c^	19.56	16
**Glu**	109.60 ± 4.73 ^g^	133.19 ± 7.69 ^f^	143.16± 1.87 ^def^	16.24 ± 0.47 ^abc^	175.41 ± 4.96 ^ab^	176.24 ± 2.19 ^a^	161.60 ± 8.24 ^abc^	158.48 ± 7.69 ^bcd^	174.43 ± 3.61 ^ab^	150.76 ± 7.46 ^cde^	145.5 8± 0.89 ^cdef^	139.28 ± 6.16 ^ef^	133.42 ± 1.95 ^f^	66.64	10
**Pro**	73.95 ± 3.41 ^cd^	68.84 ± 2.81 ^d^	74.54 ± 2.33 ^cd^	79.52 ± 3.94 ^cd^	89.80 ± 5.34 ^bc^	101.18 ± 8.00 ^ab^	101.38 ± 7.05 ^ab^	98.32 ± 4.60 ^ab^	107.03 ± 9.79 ^ab^	110.28 ± 1.77 ^a^	108.34 ± 4.18 ^ab^	102.92 ± 9.83 ^ab^	107.88 ± 1.54 ^ab^	36.33	18
**Arg**	141.29 ± 3.16 ^b^	156.59 ± 6.03 ^a^	147.95 ± 3.15 ^ab^	150.39 ± 6.79 ^ab^	139.18 ± 6.34 ^b^	83.14 ± 7.84 ^c^	65.65 ± 1.60 ^d^	55.41 ± 1.87 ^d^	0.00 ± 0.00 ^e^	0.00 ± 0.00 ^e^	0.00 ± 0.00 ^e^	0.00 ± 0.00 ^e^	0.00 ± 0.00 ^e^	15.31	2
**Ala**	114.10 ± 1.08 ^a^	123.10 ± 6.82 ^a^	125.31 ± 3.44 ^a^	128.66 ± 7.56 ^a^	117.87 ± 5.62 ^a^	90.83 ± 6.05 ^a^	87.06 ± 6.78 ^b^	86.58 ± 5.53 ^b^	91.88 ± 5.98 ^b^	83.37 ± 7.70 ^b^	81.82 ± 0.21 ^b^	76.79 ± 2.54 ^b^	78.77 ± 2.13 ^b^	14.56	6
**Val**	69.79 ± 0.84 ^bc^	72.43 ± 4.78 ^bc^	68.45 ± 4.76 ^c^	84.83 ± 8.13 ^abc^	94.68 ± 6.71 ^a^	87.23 ± 9.50 ^abc^	92.78 ± 8.68 ^a^	80.43 ± 4.25 ^abc^	88.61 ± 4.64 ^ab^	83.23 ± 7.81 ^abc^	88.72 ± 2.56 ^ab^	82.88 ± 1.07 ^abc^	81.16 ± 1.15 ^abc^	24.92	8
**Leu**	223.27 ± 9.04 ^e^	274.81 ± 3.48 ^bc^	269.09 ± 8.90 ^bc^	262.78 ± 8.73 ^bc^	277.75 ± 5.61 ^bc^	285.27 ± 6.25 ^ab^	302.91 ± 7.02 ^a^	302.62 ± 9.85 ^a^	301.61 ± 8.67 ^a^	254.14 ± 9.35 ^cd^	234.91 ± 9.63 ^de^	219.65 ± 3.95 ^e^	225.93 ± 5.83 ^e^	79.64	12
**Phe**	97.42 ± 0.23 ^bcd^	108.24 ± 8.07 ^ab^	108.10 ± 7.32 ^ab^	108.48 ± 9.44 ^ab^	110.57 ± 9.43 ^ab^	117.38 ± 3.08 ^a^	115.54 ± 6.00 ^ab^	103.55 ± 2.89 ^ab^	102.59 ± 3.47 ^ab^	85.64 ± 4.64 ^cde^	80.84 ± 4.23 ^de^	75.89 ± 1.21 ^e^	77.26 ± 2.07 ^e^	19.96	10
**Trp**	84.93 ± 3.59 ^ab^	84.61 ± 1.83 ^ab^	85.82 ± 1.36 ^a^	81.61 ± 5.27 ^ab^	84.30 ± 1.39 ^ab^	80.23 ± 0.64 ^ab^	52.31 ± 2.26 ^b^	0.00 ± 0.00 ^c^	0.00 ± 0.00 ^c^	0.00 ± 0.00 ^c^	0.00 ± 0.00 ^c^	0.00 ± 0.00 ^c^	0.00 ± 0.00 ^c^	0.90	4
**Tyr**	79.29 ± 2.54 ^a^	78.88 ± 0.35 ^ab^	79.37 ± 1.47 ^a^	77.60 ± 7.73 ^ab^	74.00 ± 5.74 ^abcd^	75.49 ± 2.52 ^abc^	74.11 ± 2.59 ^abcd^	77.91 ± 4.83 ^ab^	76.21 ± 2.57 ^abc^	66.93 ± 5.01 ^bcde^	65.18 ± 1.46 ^cde^	63.07 ± 0.91 ^de^	59.45 ± 1.41 ^e^	0.09	4
**Ser**	39.85 ± 2.08 ^b^	50.8 ± 5.40 ^ab^	55.59 ± 2.22 ^a^	52.74 ± 7.98 ^a^	50.01 ± 3.96 ^ab^	51.37 ± 3.48 ^ab^	47.88 ± 2.73 ^ab^	49.45 ± 2.68 ^ab^	53.45 ± 3.14 ^a^	47.44 ± 5.11 ^ab^	45.96 ± 0.82 ^ab^	43.04 ± 1.70 ^ab^	46.91 ± 0.50 ^ab^	15.74	4
**Gly**	35.07 ± 2.23 ^d^	42.47 ± 3.26 ^cd^	43.12 ± 1.24 ^cd^	51.84 ± 9.00 ^abc^	56.02 ± 5.56 ^abc^	54.89 ± 4.55 ^abc^	55.57 ± 3.64 ^abc^	57.41± 2.93 ^ab^	64.94 ± 3.31 ^a^	55.60 ± 6.30 ^abc^	53.85 ± 0.19 ^abc^	47.27 ± 2.35 ^bcd^	51.46 ± 0.79 ^abc^	29.87	16
**Cy2**	120.10 ± 7.89 ^cde^	127.24 ± 3.26 ^cd^	107.54 ± 2.98 ^e^	123.89 ± 8.17 ^cde^	131.89 ± 5.12 ^bcd^	127.89 ± 5.00 ^bcd^	134.96 ± 5.79 ^abc^	126.14 ± 2.36 ^cd^	151.49 ± 9.58 ^a^	137.09 ± 8.00 ^abc^	146.08 ± 2.14 ^ab^	113.83 ± 3.54 ^de^	127.56 ± 2.89 ^bcd^	31.39	16

Note: Glutamine and histidine were not detected. Asp, aspartate; Asn, asparagine; Lys, lysine; Met, methionine; Ile, isoleucine; Glu, glutamate; Pro, proline; Arg, arginine; Ala, alanine; Val, valine; Leu, leucine; Phe, phenylalanine; Trp, tryptophan; Tyr, tyrosine; Ser, serine; Gly, glycine; Cy2, cystine. Values are means ± standard error of the mean (SEM), *n* = 3. Mean ± SEM within the same row sharing a common superscript letter are not significantly different at *p* > 0.05.

**Table 6 ijms-20-01777-t006:** The amino acid production profile of *P. acidilactici* UL-3.

AA (mg/L)	Incubation Time (h)													Maximum Increment	
	0	2	4	6	8	10	12	14	16	18	20	22	24	Amount (mg/L)	Time (h)
**Asp**	62.02 ± 1.33 ^a^	61.29 ± 3.42 ^a^	60.27 ± 2.06 ^a^	61.23 ± 0.89 ^a^	61.56 ± 2.36 ^a^	62.11 ± 7.02 ^a^	61.81 ± 2.51 ^a^	62.79 ± 3.85 ^a^	60.95 ± 2.39 ^a^	62.13 ± 0.95 ^a^	60.65 ± 2.95 ^a^	60.35 ± 0.94 ^a^	60.24 ± 2.10 ^a^	0.77	14
**Asn**	76.51 ± 3.89 ^a^	77.15 ± 2.63 ^a^	75.96 ± 1.67 ^a^	75.72 ± 1.47 ^a^	76.33 ± 2.28 ^a^	75.74 ± 8.06 ^a^	76.98 ± 2.33 ^a^	76.52 ± 4.95 ^a^	76.14 ± 6.90 ^a^	76.60 ± 0.72 ^a^	75.24 ± 5.43 ^a^	76.17 ± 1.81 ^a^	76.40 ± 0.77 ^a^	0.64	2
**Lys**	367.47 ± 7.30 ^a^	340.81 ± 10.05 ^ab^	328.24 ± 1.66 ^b^	336.36 ± 9.21 ^ab^	323.94 ± 5.35 ^b^	311.66 ± 10.51 ^b^	320.94 ± 7.37 ^b^	315.93 ± 4.08 ^b^	316.68 ± 3.82 ^b^	320.71 ± 5.49 ^b^	311.92 ± 34.43 ^b^	318.66 ± 7.01 ^b^	328.29 ± 4.99 ^b^	0.00	0
**Thr**	70.78 ± 1.11 ^a^	70.77 ± 2.33 ^a^	64.77 ± 3.75 ^ab^	61.15 ± 2.72 ^abc^	65.38 ± 2.40 ^ab^	61.81 ± 6.89 ^abc^	57.20 ± 2.52 ^bc^	5.7.84 ± 1.11 ^bc^	57.05 ± 3.75 ^bc^	58.11 ± 2.40 ^bc^	53.31 ± 2.51 ^c^	59.99 ± 1.08 ^bc^	58.28 ± 0.40 ^bc^	0.00	0
**Met**	106.52 ± 0.68 ^c^	111.07 ± 2.97 ^bc^	105.76 ± 1.52 ^c^	116.59 ± 4.07 ^abc^	119.22 ± 7.06 ^ab^	120.48 ± 7.60 ^ab^	124.61 ± 0.42 ^a^	126.33 ± 3.95 ^a^	125.74 ± 4.46 ^a^	12.8.45 ± 2.63 ^a^	126.58 ± 1.67 ^a^	128.03 ± 3.22 ^a^	129.72 ± 3.87 ^a^	23.21	24
**Ile**	99.15 ± 4.71 ^f^	116.86 ± 3.65 ^e^	120.81 ± 1.22 ^de^	133.82 ± 3.33 ^bc^	150.22 ± 1.30 ^a^	145.91 ± 4.15 ^a^	149.50 ± 0.99 ^a^	142.32 ± 1.55 ^ab^	125.08 ± 2.39 ^cde^	126.95 ± 6.94 ^cde^	121.33 ± 5.38 ^de^	129.83 ± 5.60 ^cd^	143.38 ± 2.12 ^ab^	51.07	8
**Glu**	228.52 ± 4.89 ^cd^	234.65 ± 8.34 ^bcd^	225.80 ± 8.34 ^d^	237.21 ± 8.94 ^bcd^	254.74 ± 8.18 ^abcd^	260.23 ± 24.95 ^ab^	269.48 ± 4.44 ^a^	256.69 ± 4.85 ^abcd^	249.90 ± 8.93 ^abcd^	251.99 ± 3.29 ^abcd^	245.09 ± 0.50 ^abcd^	248.68 ± 6.62 ^abcd^	258.82 ± 1.18 ^abc^	40.96	12
**Pro**	50.25 ± 1.12 ^g^	50.52 ± 0.21 ^g^	52.03 ± 0.34 ^g^	52.75 ± 0.40 ^g^	62.31 ± 0.70 ^f^	67.25 ± 0.45 ^e^	71.58 ± 0.92 ^d^	70.91 ± 0.26 ^d^	74.10 ± 1.21 ^c^	77.91 ± 0.77 ^b^	79.29 ± 0.49 ^b^	82.08 ± 1.97 ^a^	81.95 ± 0.51 ^a^	31.83	22
**Arg**	186.26 ± 6.63 ^a^	177.25 ± 7.49 ^ab^	177.56 ± 1.55 ^ab^	178.02 ± 17.62 ^ab^	170.27 ± 10.76 ^ab^	151.44 ± 10.96 ^bc^	133.47 ± 2.92 ^c^	28.1 ± 28.1 ^d^	0.00 ± 0.00 ^d^	0.00 ± 0.00 ^d^	0.00 ± 0.00 ^d^	0.00 ± 0.00 ^d^	0.00 ± 0.00 ^d^	0.00	0
**Ala**	193.89 ± 3.22 ^ab^	188.45 ± 5.76 ^b^	186.47 ± 3.06 ^b^	188.76 ± 3.15 ^b^	193.35 ± 6.54 ^b^	199.21 ± 8.51 ^ab^	199.89 ± 2.79 ^ab^	198.50 ± 3.67 ^ab^	195.47 ± 8.94 ^ab^	189.69 ± 1.63 ^b^	195.45 ± 0.80 ^ab^	202.00 ± 3.28 ^ab^	209.46 ± 1.00 ^a^	15.57	24
**Val**	78.85 ± 1.65 ^c^	84.08 ± 0.85 ^c^	82.82 ± 4.24 ^c^	91.75 ± 3.78 ^b^	98.16 ± 3.02 ^ab^	98.05 ± 3.47 ^ab^	102.03 ± 1.70 ^a^	97.27 ± 1.41 ^ab^	95.78 ± 3.10 ^ab^	100.75 ± 2.23 ^a^	95.81 ± 0.79 ^ab^	99.20 ± 3.51 ^ab^	103.39 ± 1.00 ^a^	24.54	24
**Leu**	361.65± 26.99 ^e^	406.78 ± 5.18 ^d^	409.97 ± 7.52 ^cd^	415.55 ± 8.64 ^bcd^	422.73 ± 3.19 ^abcd^	442.52 ± 6.54 ^abc^	449.45 ± 6.65 ^a^	444.06 ± 6.79 ^ab^	417.34 ± 9.89 ^abcd^	421.27 ± 6.03 ^abcd^	439.83 ± 8.29 ^abc^	426.01 ± 3.08 ^abcd^	435.13 ± 1.37 ^abcd^	87.80	12
**Phe**	121.39 ± 4.97 ^a^	115.87 ± 3.79 ^ab^	112.87 ± 0.62 ^abc^	118.34 ± 4.65 ^a^	115.55 ± 0.20 ^ab^	105.51 ± 0.37 ^bc^	110.20 ± 4.18 ^abc^	103.73 ± 4.66 ^c^	92.51 ± 2.10 ^d^	93.31 ± 4.90 ^d^	90.56 ± 5.22 ^d^	91.50 ± 1.78 ^d^	89.74 ± 1.10 ^d^	0.00	0
**Tyr**	81.40 ± 1.31 ^ab^	77.76 ± 2.18 ^b^	77.20 ± 2.41 ^b^	77.07 ± 1.19 ^b^	64.45 ± 32.63 ^a^	60.86 ± 32.53 ^b^	57.38 ± 28.76 ^b^	51.42 ± 25.71 ^b^	25.78 ± 25.78 ^c^	0.00 ± 0.00 ^c^	0.00 ± 0.00 ^c^	0.00 ± 0.00 ^c^	0.00 ± 0.00 ^c^	0.00	0
**Trp**	123.86 ± 4.74 ^a^	116.10 ± 2.91 ^abc^	116.42 ± 1.72 ^abc^	107.48 ± 2.26 ^c^	120.78 ± 6.55 ^ab^	83.15 ± 44.36 ^bc^	0.00 ± 0.00 ^d^	0.00 ± 0.00 ^d^	0.00 ± 0.00 ^d^	0.00 ± 0.00 ^d^	0.00 ± 0.00 ^d^	0.00 ± 0.00 ^d^	0.00 ± 0.00 ^d^	0.00	0
**Ser**	115.91 ± 1.46 ^de^	114.70 ± 4.83 ^e^	112.19 ± 1.6 ^e^	119.61 ± 3.2 ^de^	128.27± 6.59 ^cde^	125.69 ± 9.25 ^cde^	135.37 ± 4.76 ^bcd^	129.04 ± 4.95 ^cde^	127.03 ± 7.07 ^cde^	131.53 ± 1.75 ^cde^	144.44 ± 8.05 ^abc^	150.31 ± 8.74 ^ab^	161.43 ± 6.28 ^a^	45.52	24
**Gly**	89.67 ± 5.90 ^de^	85.66 ± 4.57 ^e^	94.94 ± 2.01 ^de^	90.33 ± 4.15 ^de^	98.44 ± 8.06 ^cde^	97.25 ± 8.53 ^cde^	108.25 ± 8.25 ^bcd^	104.17 ± 5.06 ^bcde^	108.72 ± 4.19 ^bcd^	116.42 ± 5.98 ^bc^	116.66 ± 4.46 ^bc^	135.69 ± 6.59 ^a^	121.28 ± 5.93 ^ab^	46.02	22
**Cy2**	163.78 ± 4.44 ^ab^	159.38 ± 6.14 ^ab^	153.75 ± 10.34 ^b^	158.12 ± 3.27 ^ab^	168.81 ± 11.24 ^ab^	175.34 ± 21.02 ^ab^	184.80 ± 8.32 ^ab^	179.93 ± 7.02 ^ab^	193.40 ± 18.37 ^ab^	186.42 ± 16.46 ^ab^	193.74 ± 20.43 ^ab^	204.47 ± 14.54 ^a^	201.38 ± 18.08 ^a^	40.69	22

Note: Glutamine and histidine were not detected. Asp, aspartate; Asn, asparagine; Lys, lysine; Met, methionine; Ile, isoleucine; Glu, glutamate; Pro, proline; Arg, arginine; Ala, alanine; Val, valine; Leu, leucine; Phe, phenylalanine; Trp, tryptophan; Tyr, tyrosine; Ser, serine; Gly, glycine; Cy2, cystine. Values are means ± standard error of the mean (SEM), *n* = 3. Mean ± SEM within the same row sharing a common superscript letter are not significantly different at *p* > 0.05.

**Table 7 ijms-20-01777-t007:** The amino acid production profile of *L. plantarum* UL-4.

AA (mg/L)	Incubation Time (h)													Maximum Increment	
	0	2	4	6	8	10	12	14	16	18	20	22	24	Amount (mg/L)	Time (h)
**Asp**	40.89 ± 0.29 ^b^	40.57 ± 1.91 ^b^	41.61 ± 1.26 ^b^	47.47 ± 1.12 ^a^	39.66 ± 0.46 ^b^	35.93 ± 0.46 ^c^	32.47 ± 1.00 ^d^	0.00 ± 0.00 ^e^	0.00 ± 0.00 ^e^	0.00 ± 0.00 ^e^	0.00 ± 0.00 ^e^	0.00 ± 0.00 ^e^	0.00 ± 0.00 ^e^	6.58	6
**Asn**	48.90 ± 0.40 ^b^	50.20 ± 0.59 ^a^	40.99 ± 0.03 ^c^	49.15 ± 0.68 ^b^	0.00 ± 0.00 ^d^	0.00 ± 0.00 ^d^	0.00 ± 0.00 ^d^	0.00 ± 0.00 ^d^	0.00 ± 0.00 ^d^	0.00 ± 0.00 ^d^	0.00 ± 0.00 ^d^	0.00 ± 0.00 ^d^	0.00 ± 0.00 ^d^	1.30	2
**Lys**	357.18 ± 1.82 ^a^	350.95 ± 8.79 ^ab^	351.82 ± 4.79 ^ab^	359.05 ± 16.93 ^a^	343.97 ± 2.91 ^abc^	328.77 ± 5.03 ^bcd^	333.41 ± 3.81 ^bc^	325.66 ± 10.47 ^cd^	321.30 ± 1.45 ^cde^	324.01 ± 1.37 ^cde^	305.18 ± 6.89 ^def^	295.97 ± 9.95 ^f^	301.17 ± 3.05 ^ef^	1.87	6
**Met**	91.27 ± 0.66 ^c^	93.79 ± 2.93 ^bc^	94.83 ± 1.74 ^abc^	95.83 ± 1.82 ^abc^	99.87 ± 1.44 ^a^	96.93 ± 2.06 ^abc^	97.81 ± 1.26 ^ab^	95.55 ± 2.36 ^abc^	96.10 ± 0.89 ^abc^	94.69 ± 1.05 ^abc^	91.30 ± 0.03 ^c^	91.61± 0.87 ^c^	91.60 ± 2.34 ^c^	8.61	8
**Thr**	66.37 ± 3.39 ^b^	62.55 ± 5.40 ^b^	63.53 ± 4.71 ^b^	83.05 ± 2.25 ^a^	65.15 ± 6.01 ^b^	63.78 ± 2.52 ^b^	57.71 ± 4.21 ^bc^	47.71 ± 1.70 ^cd^	47.7 ± 0.78 ^cd^	43.47 ± 1.19 ^d^	41.79 ± 1.47 ^d^	38.97 ± 0.93 ^d^	38.52 ± 2.77 ^d^	16.67	6
**Ile**	87.50 ± 0.90 ^g^	89.89 ± 1.87 ^fg^	92.77 ± 0.07 ^f^	112.95 ± 0.73 ^a^	103.58 ± 2.16 ^bc^	102.47 ± 1.85 ^cd^	107.08 ± 1.45 ^b^	101.0 ± 0.92 ^cde^	103.8 ± 1.47 ^bc^	101.9 ± 1.48 ^d^	98.3 ± 1.38 ^de^	97.3 ± 1.18 ^e^	98.7 ± 1.32 ^de^	25.45	6
**Glu**	215.74 ± 1.50 ^g^	215.52 ± 5.70 ^g^	220.18 ± 2.23 ^fg^	253.11 ± 3.51 ^a^	248.47 ± 6.16 ^ab^	243.75 ± 4.89 ^abc^	248.81 ± 3.49 ^ab^	238.37 ± 25.1 ^bcde^	241.0 ± 1.85 ^bcd^	232.9 ± 2.58 ^cde^	228.9 ± 5.13 ^ef^	229.5 ± 1.60 ^def^	227.2 ± 2.28 ^ef^	37.38	6
**Pro**	36.49 ± 0.59 ^c^	36.72 ± 0.11 ^c^	35.95 ± 0.53 ^c^	37.27 ± 3.06 ^c^	37.78 ± 1.47 ^c^	39.37 ± 1.12 ^bc^	45.46 ± 0.74 ^a^	42.75 ± 1.15 ^ab^	43.02 ± 1.25 ^ab^	44.74 ± 0.93 ^a^	43.90 ± 1.62 ^a^	44.38 ± 1.50 ^a^	42.55 ± 0.78 ^ab^	8.97	12
**Arg**	203.69 ± 1.60 ^abc^	208.44 ± 1.26 ^ab^	210.51 ± 2.14 ^a^	203.07± 6.41 ^abc^	200.21 ± 2.81 ^bcd^	192.06 ± 2.98 ^def^	195.50 ± 1.51 ^cde^	193.80 ± 3.01 ^cde^	195.70 ± 1.98 ^cde^	189.61 ± 1.05 ^ef^	176.24 ± 6.19 ^h^	173.33 ± 2.99 ^h^	183.62 ± 0.78 ^fg^	6.81	4
**Ala**	165.93 ± 1.56 ^b^	165.34 ± 1.21 ^b^	165.65 ± 0.98 ^b^	176.29 ± 2.07 ^a^	162.34 ± 0.84 ^b^	147.74 ± 2.27 ^c^	145.05 ± 2.55 ^cd^	137.86 ± 1.55 ^d^	136.66 ± 1.08 ^d^	125.33 ± 1.79 ^e^	116.43 ± 7.23 ^f^	117.07 ± 4.67 ^f^	115.21 ± 1.54 ^f^	10.37	6
**Val**	111.28 ± 0.91 ^d^	112.01 ± 1.76 ^d^	113.96 ± 1.70 ^d^	126.04± 1.10 ^abc^	125.31 ± 1.18 ^bc^	124.29 ± 2.62 ^bc^	131.39 ± 2.00 ^a^	127.34 ± 2.61 ^ab^	126.14± 0.37 ^abc^	122.83 ± 2.81 ^bc^	122.15 ± 1.05 ^bc^	120.30 ± 1.59 ^c^	121.16 ± 2.26 ^c^	20.11	12
**Leu**	346.55 ± 5.00 ^bcde^	349.15 ± 4.25 ^bcd^	356.35 ± 0.32 ^ab^	365.34 ± 6.26 ^a^	351.18 ± 2.97 ^bc^	336.63 ± 2.35 ^cdef^	339.97 ± 4.10 ^cdef^	332.21 ± 2.11 ^efg^	336.02 ± 4.34 ^def^	329.73 ± 3.58 ^fgh^	318.26 ± 8.97 ^gh^	315.71 ± 5.90 ^h^	321.26 ± 2.66 ^gh^	18.79	6
**Phe**	159.89 ± 0.91 ^ab^	160.54 ± 1.36 ^b^	163.07 ± 0.80 ^a^	154.53 ± 4.45 ^bc^	154.95 ± 3.52 ^bc^	148.13 ± 2.02 ^de^	152.58 ± 2.21 ^cd^	147.77 ± 0.35 ^de^	145.02 ± 0.86 ^e^	143.72 ± 0.80 ^e^	137.28 ± 1.93 ^f^	136.25 ± 1.17 ^f^	136.85 ± 2.05 ^f^	3.21	4
**Trp**	80.23 ± 00.67 ^a^	77.49 ± 0.68 ^b^	77.21 ± 1.69 ^b^	78.35 ± 1.55 ^ab^	72.37 ± 0.26 ^c^	70.36 ± 0.49 ^c^	72.02 ± 0.88 ^c^	0.00 ± 0.00 ^d^	0.00 ± 0.00 ^d^	0.00 ± 0.00 ^d^	0.00 ± 0.00 ^d^	0.00 ± 0.00 ^d^	0.00 ± 0.00 ^d^	0.00	0
**Tyr**	65.84 ± 0.41 ^a^	62.03 ± 0.58 ^abc^	64.05 ± 0.55 ^ab^	55.23 ± 3.89 ^def^	62.28 ± 1.21 ^abc^	57.77 ± 0.99 ^cd^	56.92 ± 1.12 ^cde^	55.88 ± 1.25 ^de^	58.14 ± 0.74 ^bcd^	53.38 ± 0.97 ^def^	51.00 ± 3.52 ^ef^	49.68 ± 3.35 ^f^	51.09 ± 0.45 ^ef^	0.00	0
**Ser**	114.44 ± 1.49 ^b^	113.36 ± 0.16 ^b^	113.00 ± 0.35 ^b^	139.32 ± 3.04 ^a^	0.00 ± 0.00 ^c^	0.00 ± 0.00 ^c^	0.00 ± 0.00 ^c^	0.00 ± 0.00 ^c^	0.00 ± 0.00 ^c^	0.00 ± 0.00 ^c^	0.00 ± 0.00 ^c^	0.00 ± 0.00 ^c^	0.00 ± 0.00 ^c^	24.88	6
**Gly**	67.04 ± 0.90 ^cde^	63.40 ± 0.33 ^e^	64.15 ± 0.16 ^de^	78.20 ± 2.95 ^ab^	71.15 ± 0.30 ^de^	73.11 ± 0.70 ^abcde^	76.37 ± 0.95 ^abc^	77.07 ± 1.38 ^abc^	82.19 ± 2.46 ^a^	76.98 ± 2.50 ^abc^	73.99 ± 7.97 ^abcd^	75.49 ± 5.15 ^abc^	78.38 ± 2.58 ^ab^	15.15	16
**Cy2**	128.81 ± 5.42 ^a^	100.84 ± 3.63 ^bc^	101.21 ± 5.18 ^bc^	113.18 ± 2.96 ^ab^	93.72 ± 0.08 ^bcd^	87.01± 1.97 ^cde^	84.13 ± 3.50 ^cde^	74.43 ± 2.45 ^de^	87.54 ± 1.46 ^cde^	78.56 ± 11.14 ^de^	72.66 ± 9.04 ^de^	69.01 ± 16.07 ^e^	76.44 ± 2.62 ^de^	0.00	0

Note: Glutamine and histidine were not detected. Asp, aspartate; Asn, asparagine; Lys, lysine; Met, methionine; Ile, isoleucine; Glu, glutamate; Pro, proline; Arg, arginine; Ala, alanine; Val, valine; Leu, leucine; Phe, phenylalanine; Trp, tryptophan; Tyr, tyrosine; Ser, serine; Gly, glycine; Cy2, cystine. Values are means ± standard error of the mean (SEM), *n* = 3. Mean ± SEM within the same row sharing a common superscript letter are not significantly different at *p* > 0.05.

**Table 8 ijms-20-01777-t008:** The amino acid production profile of *P. pentosaceus* UL-6.

AA (mg/L)	Incubation Time (h)													Maximum Increment	
	0	2	4	6	8	10	12	14	16	18	20	22	24	Amount (mg/L)	Time (h)
**Asp**	47.12 ± 3.94 ^a^	46.56 ± 6.06 ^a^	45.80 ± 2.38 ^a^	46.37 ± 3.48 ^a^	47.64 ± 5.93 ^a^	46.88 ± 7.20 ^a^	46.23 ± 6.17 ^a^	47.92 ± 5.75 ^a^	45.75 ± 3.70 ^a^	47.64 ± 7.27 ^a^	46.58 ± 5.69 ^a^	48.90 ± 2.22 ^a^	45.43 ± 4.88 ^a^	1.79	22
**Asn**	53.18 ± 4.44 ^a^	49.43 ± 4.11 ^a^	55.91 ± 3.12 ^a^	51.47 ± 5.69 ^a^	56.47 ± 4.61 ^a^	50.82 ± 7.45 ^a^	51.27 ± 4.47 ^a^	52.78 ± 8.23 ^a^	50.03 ± 5.57 ^a^	57.47 ± 9.13 ^a^	57.41 ± 8.16 ^a^	58.98 ± 2.70 ^a^	52.40 ± 7.12 ^a^	5.80	22
**Lys**	369.72 ± 7.13 ^a^	311.54 ± 1.07 ^b^	307.32 ± 7.88 ^b^	275.70 ± 7.99 ^c^	270.01 ± 5.27 ^c^	240.95 ± 4.78 ^d^	272.85 ± 5.47 ^c^	235.75 ± 5.25 ^d^	203.09 ± 8.77 ^e^	236.22 ± 4.36 ^d^	239.71 ± 3.74 ^d^	252.87 ± 2.55 ^d^	236.28 ± 6.19 ^d^	0.00	0
**Met**	97.00 ± 2.02 ^ab^	96.03 ± 8.26 ^ab^	100.09 ± 8.74 ^ab^	102.82 ± 3.50 ^ab^	111.37 ± 7.46 ^a^	101.72 ± 3.09 ^ab^	108.77 ± 12.08 ^a^	106.36 ± 3.31 ^a^	83.41 ± 0.58 ^b^	59.69 ± 8.51 ^c^	59.20 ± 8.47 ^c^	60.78 ± 1.62 ^c^	54.33 ± 2.51 ^c^	14.37	8
**Thr**	71.61 ± 4.82 ^a^	70.04 ± 6.00 ^a^	70.70± 8.20 ^a^	71.24 ± 7.18 ^a^	66.06 ± 8.86 ^ab^	65.60 ± 5.35 ^ab^	61.36 ± 1.20 ^ab^	62.82 ± 3.09 ^ab^	59.46 ± 0.68 ^ab^	55.39 ± 9.26 ^ab^	55.10 ± 4.28 ^ab^	52.68 ± 5.25 ^ab^	48.93 ± 3.91 ^b^	0.00	0
**Ile**	113.02 ± 2.43 ^bc^	99.46 ± 7.20 ^c^	102.88 ± 4.70 ^c^	99.30 ± 8.63 ^c^	104.65 ± 6.45 ^c^	126.34 ± 1.42 ^ab^	129.62 ± 4.81 ^a^	102.43 ± 4.83 ^c^	78.81 ± 2.42 ^d^	72.54 ± 7.05 ^d^	75.60 ± 4.79 ^d^	74.53 ± 1.43 ^d^	67.85 ± 2.03 ^d^	16.60	12
**Glu**	205.26 ± 2.95 ^cde^	194.95 ± 5.30 ^e^	205.46± 9.36 ^cde^	234.70 ± 6.66 ^ab^	244.13 ± 9.57 ^a^	200.55 ± 3.22 ^de^	230.05 ± 9.16 ^ab^	216.88 ± 4.14 ^bcd^	203.30 ± 7.06 ^cde^	222.50 ± 5.17 ^bc^	226.22 ± 2.93 ^ab^	216.45± 6.08 ^bcd^	191.70 ± 4.15 ^e^	38.87	8
**Pro**	47.65 ± 2.36 ^d^	46.02 ± 2.03 ^d^	45.23 ± 3.80 ^d^	46.34 ± 1.08 ^d^	47.41 ± 1.28 ^d^	56.14 ± 7.79 ^d^	60.25 ± 6.48 ^d^	122.09 ± 3.86 ^c^	125.27 ± 6.64 ^c^	149.27 ± 4.72 ^b^	158.85 ± 5.05 ^ab^	163.71 ± 8.46 ^ab^	169.53 ± 6.47 ^a^	121.88	24
**Arg**	226.39 ± 3.50 ^a^	216.13 ± 4.53 ^b^	208.92 ± 7.25 ^b^	164.05 ± 5.45 ^c^	118.73 ± 5.21 ^d^	0.00 ± 0.00 ^e^	0.00 ± 0.00 ^e^	0.00 ± 0.00 ^e^	0.00 ± 0.00 ^e^	0.00 ± 0.00 ^e^	0.00 ± 0.00 ^e^	0.00 ± 0.00 ^e^	0.00 ± 0.00 ^e^	0.00	0
**Ala**	176.49 ± 8.07 ^a^	165.51 ± 6.97 ^a^	156.90 ± 7.09 ^a^	157.70 ± 5.94 ^a^	156.33 ± 6.14 ^a^	119.49 ± 9.41 ^bc^	134.15 ± 5.89 ^b^	123.49 ± 9.01 ^bc^	106.18 ± 5.22 ^cd^	125.95 ± 5.36 ^bc^	130.30 ± 6.51 ^b^	120.63 ± 0.87 ^bc^	92.74 ± 4.52 ^d^	0.00	0
**Val**	56.46 ± 2.33 ^cd^	53.42 ± 6.11 ^d^	59.24 ± 8.72 ^bcd^	63.58 ± 1.92 ^abcd^	81.39 ± 3.76 ^a^	64.53 ± 6.78 ^abcd^	62.98 ± 1.45 ^bcd^	68.59 ± 7.37 ^abcd^	67.50 ± 3.64 ^abcd^	74.98 ± 9.06 ^ab^	74.44 ± 3.57 ^abc^	76.13 ± 5.80 ^ab^	75.82 ± 1.14 ^ab^	24.93	8
**Leu**	392.77 ± 1.94 ^a^	344.25 ± 3.80 ^b^	330.83 ± 7.26 ^bc^	301.26 ± 6.73 ^d^	295.62 ± 9.41 ^d^	260.28 ± 4.88 ^e^	256.08 ± 7.71 ^e^	267.76 ± 7.25 ^e^	254.69 ± 3.47 ^e^	255.55 ± 6.34 ^e^	269.14 ± 9.42 ^e^	313.59 ± 9.04 ^cd^	270.71 ± 5.85 ^e^	0.00	0
**Phe**	84.74 ± 2.38 ^cde^	76.96 ± 8.91 ^e^	84.15 ± 4.09 ^de^	99.94 ± 7.23 ^bcd^	99.46 ± 4.53 ^bcd^	113.35 ± 8.62 ^ab^	124.66 ± 3.68 ^a^	102.93 ± 6.92 ^bcd^	90.41 ± 4.16 ^cde^	104.67 ± 9.03 ^bc^	101.71 ± 3.96 ^bcd^	102.26 ± 3.02 ^bcd^	97.32 ± 5.60 ^bcd^	39.92	12
**Trp**	83.33 ± 6.13 ^a^	82.09 ± 4.04 ^a^	86.71 ± 3.29 ^a^	85.54 ± 7.92 ^a^	82.30 ± 12.60 ^a^	61.62 ± 0.96 ^a^	56.67 ± 6.22 ^a^	0.00 ± 0.00 ^b^	0.00 ± 0.00 ^b^	0.00 ± 0.00 ^b^	0.00 ± 0.00 ^b^	0.00 ± 0.00 ^b^	0.00 ± 0.00 ^b^	3.39	4
**Tyr**	79.4 ± 5.71 ^a^	66.45 ± 9.00 ^b^	75.92 ± 0.47 ^b^	78.36 ± 2.49 ^b^	78.12 ± 2.68 ^b^	73.46 ± 2.71 ^b^	76.19 ± 2.76 ^b^	77.74 ± 1.06 ^b^	73.35 ± 1.05 ^b^	77.61 ± 2.93 ^b^	77.06 ± 4.57 ^b^	76.48 ± 1.32 ^b^	70.95 ± 4.89 ^ab^	0.00	0
**Ser**	110.30 ± 5.27 ^a^	90.27 ± 9.41 ^ab^	93.66 ± 8.79 ^ab^	89.92 ± 7.66 ^ab^	89.72 ± 5.42 ^ab^	62.44 ± 8.25 ^c^	83.77 ± 6.90 ^b^	60.53 ± 9.00 ^c^	55.75 ± 2.37 ^c^	63.20 ± 2.11 ^c^	61.84 ± 4.82 ^c^	54.90 ± 2.38 ^c^	54.95 ± 6.49 ^c^	0.00	0
**Gly**	62.36 ± 5.0 ^d^	93.92 ± 5.72 ^c^	94.10 ± 6.73 ^c^	122.11 ± 2.40 ^ab^	133.42 ± 6.17 ^a^	111.73 ± 6.88 ^bc^	108.26 ± 8.35 ^bc^	107.08 ± 4.73 ^bc^	100.14 ± 8.62 ^c^	99.12 ± 8.55 ^c^	74.08 ± 7.18 ^d^	68.58 ± 4.53 ^d^	60.51 ± 2.28 ^d^	71.06	8
**Cy2**	155.90 ± 2.81 ^bcd^	170.93 ± 3.80 ^ab^	160.30 ± 3.68 ^bcd^	166.23 ± 7.44 ^abc^	180.94 ± 8.66 ^a^	160.21 ± 6.26 ^bcd^	158.18 ± 9.73 ^bcd^	164.89 ± 6.88 ^abc^	136.12 ± 1.21 ^ef^	148.29 ± 1.88 ^cde^	130.73 ± 1.63 ^ef^	144.15 ± 7.63 ^de^	124.15 ± 7.82 ^f^	25.04	8

Note: Glutamine and histidine were not detected. Asp, aspartate; Asn, asparagine; Lys, lysine; Met, methionine; Ile, isoleucine; Glu, glutamate; Pro, proline; Arg, arginine; Ala, alanine; Val, valine; Leu, leucine; Phe, phenylalanine; Trp, tryptophan; Tyr, tyrosine; Ser, serine; Gly, glycine; Cy2, cystine. Values are means ± standard error of the mean (SEM), *n* = 3. Mean ± SEM within the same row sharing a common superscript letter are not significantly different at *p* > 0.05.

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
