# Peer review of "Extracellular Proteolytic Activity and Amino Acid Production by Lactic Acid Bacteria Isolated from Malaysian Foods"

_ijms, 2019, doi:10.3390/ijms20071777_

Reviewer 1 Report

   Dear Editor

This manuscript describes an extracellular proteolytic activity and amino acid production by lactic acid bacteria from foods. Although this is a carefully done and the finding are of considerable interest For the befit for the readers, however, a number of points need clarifying and certain statements require further justification. These are given below.

1)    The maximum increment amount of each amino acid together with time should be added in each Table, otherwise it seems to be inconvenient for the reader to confirm or understand the author’s claims.

 2)    L 210, P. pentosaceus UP-2 should be P. acidilactici UP-2 from the meaning of the sentences from L213 to L217 and Table 2.

3)    L289-292, This sentence is obscure, since it seems that each increment of the proteolytic activities was not so significant. As for the proteolytic activity of P. pentosaceus UB-8, this reviewer wonders why the significant increased proteolytic activity decreased rapidly after 12 hrs.

4)    L 68, Met is a chiral amino acid, Gly is only one achiral amino acid un these amino acid.

5)    L383, rReducing → Reducing. “r” should be removed.

Professor Yoshinobu Kimura, Ph D

Graduate School of Environmal and Life Science.

Okayama University, Japan.

Author Response

Response to Reviewer 1 Comments:

Comments and Suggestions for Authors:

This manuscript describes an extracellular proteolytic activity and amino acid production by lactic acid bacteria from foods. Although this is a carefully done and the finding are of considerable interest. For the benefit of readers, however, a number of points need clarifying and certain statements require further justification. These are given below.

Point 1: The maximum increment amount of each amino acid together with time should be added in each Table, otherwise it seems to be inconvenient for the reader to confirm or understand the author’s claims.

Response 1:

Table 1 to Table 8:

Maximum increment of each amino acid together with incubation time was added in each table.

Point 2: L 210, P. pentosaceus UP-2 should be P. acidilactici UP-2 from the meaning of the sentences from L213 to L217 and Table 2.

Response 2:

P. acidilactici UP-2 should be P. pentosaceus UP-2. P. acidilactici UP-2 has been corrected to P. pentosaceus UP-2 in Table 2.

Point 3:    L289-292, This sentence is obscure, since it seems that each increment of the proteolytic activities was not so significant. As for the proteolytic activity of P. pentosaceus UB-8, this reviewer wonders why the significant increased proteolytic activity decreased rapidly after 12 hrs.

Response 3:

Line 312-317 have been re-worded as follows:

“On the other hand, P. pentosaceus UB-8 showed a major increment of proteolytic activity (5.15 U/mg) at pH 6.5 between 6 to 12 h of incubation. However, proteolytic activity decreased rapidly after 12 hrs of incubation when the producer cells reached stationary phase as shown in Figure 2D. This was in line with the findings reported by Nissen-Meyer and Sletten [52], where the level of free extracellular proteases was the highest at the late exponential phase and early stationary phase of the producer cell.”

Point 4:    L 68, Met is a chiral amino acid, Gly is only one achiral amino acid un these amino acid.

Response 4:

Line 66-67 have been re-worded as follows:

“Nowadays, fermentation is widely used in the industry for the production of most AA except achiral AA such as glycine and chiral methionine, which are preferably synthesized chemically.”

Point 5:    L383, rReducing → Reducing. “r” should be removed.

Response 5:

Line 392:

“r” was removed from “Reducing”

Reviewer 2 Report

I think that the work is valid and the paper is well written. I suggest acceptance of the paper after the following minor revisions:

· The introduction is well written and it is a nice help for the reader who is not an expert of the topic. The only thing I would add in the introduction is a general discussion about the pH dependence of the proteolytic activity, including some references;

· Although the authors have been quite accurate to report all the experimental findings, I think the manuscript lacks a scheme or a table where the overall and most important results are reported, i.e. a table showing as synthetically as possible the major differences among the various LAB isolates. Analogously, a graphic illustrating the used experimental protocols would also help the reader to better understand and replicate the performed experiments.

· Lines 35-36 of the first page (abstract): please reformulate. Line 37: maybe “profound” is not the most appropriate adjective in this case.

· For future work, I think it could be nice to identify the exact experimental conditions (inhibitors, pH, etc.), which allow to select specific AA for each LAB. Do the authors think that such approach could have useful applications? Maybe a discussion on this issue could be hinted in the conclusions.

Author Response

Response to Reviewer 2 Comments:

Comments and Suggestions for Authors:

I think that the work is valid, and the paper is well written. I suggest acceptance of the paper after the following minor revisions:

Point 1: The introduction is well written, and it is a nice help for the reader who is not an expert of the topic. The only thing I would add in the introduction is a general discussion about the pH dependence of the proteolytic activity, including some references;

Response 1:

A general discussion for pH dependence of proteolytic activity with reference was included at Line 90-92:

“Besides, Gobbetti et al. [27] showed that proteinase activity was affected by pH in addition to the effects of substrate and bacterial strains.”

Point 2: Although the authors have been quite accurate to report all the experimental findings, I think the manuscript lacks a scheme or a table where the overall and most important results are reported, i.e. a table showing as synthetically as possible the major differences among the various LAB isolates.

Response 2:

Table 1 to Table 8:

Maximum increment of each amino acid together with incubation time was added in each table as suggested by reviewer 1.

Point 3: Analogously, a graphic illustrating the used experimental protocols would also help the reader to better understand and replicate the performed experiments.

Response 3:

Line 341:

Figure 4 was included to illustrate the overall experiment protocols used in the study. 

Point 4: Lines 35-36 of the first page (abstract): please reformulate.

Response 4:

Lines 35-36 was reformulated as follows:

“All studied LAB exhibited versatile extracellular proteolytic activities from acidic to alkaline pH conditions.”

Point 5: Line 37: maybe “profound” is not the most appropriate adjective in this case.

Response 5:

Line 36-40 have been reformulated as follows:

“In comparison, Pediococcus pentosaceus UP-2 exhibited the highest ability to produce 15 AA extracellularly, including aspartate, lysine, methionine, threonine, isoleucine, glutamate, proline, alanine, valine, leucine, tryptophan, tyrosine, serine, glycine and cystine, followed by Pediococcus pentosaceus UL-2, Pediococcus acidilactici UB-6 and Pediococcus acidilactici UP-1 with 11 to 12 different AA were detected extracellularly.”

Point 6: For future work, I think it could be nice to identify the exact experimental conditions (inhibitors, pH, etc.), which allow to select specific AA for each LAB. Do the authors think that such approach could have useful applications? Maybe a discussion on this issue could be hinted in the conclusions.

Response 6:

Line 422-425 have been reformulated as follows:

“It could be useful and vital information to obtain if the extracellular proteolytic enzymes of LAB can be separated and characterized further (e.g. effects of inhibitors, substrate specificity and pH, etc.) to correlate with the specific AA production by each LAB.”